# An Optimization Framework for Silicon Photonic Evanescent-Field Biosensors Using Sub-Wavelength Gratings

**DOI:** 10.3390/bios12100840

**Published:** 2022-10-08

**Authors:** Lauren S. Puumala, Samantha M. Grist, Kithmin Wickremasinghe, Mohammed A. Al-Qadasi, Sheri Jahan Chowdhury, Yifei Liu, Matthew Mitchell, Lukas Chrostowski, Sudip Shekhar, Karen C. Cheung

**Affiliations:** 1School of Biomedical Engineering, University of British Columbia, 251-2222 Health Sciences Mall, Vancouver, BC V6T 1Z3, Canada; 2Centre for Blood Research, University of British Columbia, 2350 Health Sciences Mall, Vancouver, BC V6T 1Z3, Canada; 3Dream Photonics Inc., Vancouver, BC V6T 0A7, Canada; 4Department of Electrical and Computer Engineering, University of British Columbia, 5500-2332 Main Mall, Vancouver, BC V6T 1Z4, Canada; 5Stewart Blusson Quantum Matter Institute, University of British Columbia, 2355 East Mall, Vancouver, BC V6T 1Z4, Canada

**Keywords:** silicon photonics, evanescent field biosensor, SOI biosensor, ring resonator, fishbone sub-wavelength grating waveguide, sub-wavelength grating waveguide, SWG-assist waveguide, bridged SWG waveguide, microfluidics

## Abstract

Silicon photonic (SiP) evanescent-field biosensors aim to combine the information-rich readouts offered by lab-scale diagnostics, at a significantly lower cost, and with the portability and rapid time to result offered by paper-based assays. While SiP biosensors fabricated with conventional strip waveguides can offer good sensitivity for label-free detection in some applications, there is still opportunity for improvement. Efforts have been made to design higher-sensitivity SiP sensors with alternative waveguide geometries, including sub-wavelength gratings (SWGs). However, SWG-based devices are fragile and prone to damage, limiting their suitability for scalable and portable sensing. Here, we investigate SiP microring resonator sensors designed with SWG waveguides that contain a “fishbone” and highlight the improved robustness offered by this design. We present a framework for optimizing fishbone-style SWG waveguide geometries based on numerical simulations, then experimentally measure the performance of ring resonator sensors fabricated with the optimized waveguides, targeting operation in the O-band and C-band. For the O-band and C-band devices, we report bulk sensitivities up to 349 nm/RIU and 438 nm/RIU, respectively, and intrinsic limits of detection as low as 5.1 × 10^−4^ RIU and 7.1 × 10^−4^ RIU, respectively. This performance is comparable to the state of the art in SWG-based sensors, positioning fishbone SWG resonators as an attractive, more robust, alternative to conventional SWG designs.

## 1. Introduction

The recent COVID-19 pandemic has highlighted the importance of scalable, rapid, portable, and cost-effective medical diagnostics in public safety and informed decision making [1,2]. Currently, gold-standard medical diagnostics rely on lab-based tests, which are performed in centralized settings and suffer from high costs, long analysis times, the requirement for highly trained operators, and complex logistics regarding sample transport and information management [3]. Portable, low-cost, and easy-to-use diagnostic tools, such as paper-based assays, allow for rapid and accessible testing in decentralized settings. However, they offer less information-rich readouts and often suffer from poorer sensitivity and accuracy compared with lab-based techniques [4]. Silicon photonic (SiP) biosensors offer the potential to bridge the gap between these two classes of diagnostic systems.

By leveraging highly scalable complementary metal-oxide semiconductor (CMOS) fabrication processes, SiP chips can be produced in high volumes at low cost [5,6,7,8]. Their scalability, affordability, rapid readout, and millimeter-scale form factor, makes SiP sensors amenable to testing in point-of-care (POC) settings. In addition to managing infectious diseases, rapid POC testing is valuable for the diagnosis of conditions such as stroke and sepsis, where rapid confirmation of clinical findings is critical for timely and effective treatment decision-making [9,10,11]. POC tests can also improve access to diagnostics in remote and resource-limited communities. Dozens of sensors can be fabricated on a single SiP chip, which, when combined with spatially controlled functionalization, can facilitate high-throughput multiplexed diagnostic testing [12]. This opens opportunities for more selective and information-rich diagnosis of conditions that are challenging to identify based on a single biomarker alone [9,13]. Extremely sensitive biomarker detection down to the pg/mL scale has been demonstrated on SiP platforms based on well-established strip waveguides (Figure 1c) [14,15]. However, these exceptionally low-limit of detection demonstrations have used sandwich assay formats in which the final detected signal originated from a detection antibody [16] or subsequent amplification step [14,15], rather than from the analyte itself. Label-based strategies such as these offer slower detection and require more complex assay operation than label-free formats. While label-free detection has been demonstrated with SiP platforms [17,18] and is more suitable for POC applications due to its simplicity, label-free biosensors based on strip waveguides typically have higher detection limits in the ng/mL range. Many clinical diagnostic assays require lower detection limits [19]. This has motivated the design of SiP sensors, such as microring resonators (MRRs), with improved performance criteria, including refractive index sensitivities.

MRRs use their sensitivity to surface and cladding refractive index changes to detect analytes, such as disease biomarkers, captured on the sensor surface. These MRR structures consist of a waveguide that is looped back on itself in a ring and a straight bus waveguide that couples light into the ring (Figure 1a) [20,21]. The ring and bus waveguides are separated by a defined coupling gap distance, *g_c_*, which controls the amount of light coupled into the ring (Figure 1b). Resonance occurs when the optical path length of the ring is equal to an integer multiple of the wavelength of light in the waveguide. These devices support resonances at wavelengths, λ*_res_*, given by
(1)$$\lambda_{res}=\frac{n_{eff}L}{m},$$
where *n_eff_* is the effective refractive index of the waveguide, *L* is the resonator length (L=2πR) for a circular MRR with radius, (*R*), and *m* is an integer number representing the order of interference. A portion of the electric field, called the evanescent field, travels outside of the waveguide and interacts with the surrounding material, or analyte. This creates a thin refractive index-sensitive region that extends up to a few hundred nanometers outside of the waveguide [22]. A change in the refractive index surrounding the resonator, for example due to biomolecule binding, changes the *n_eff_*, leading to a shift in λ*_res_*. Several strategies are available for tracking the resonance shifts. A simplistic setup comprises a broadband optical source that provides a continuous spectrum of wavelengths and a spectrum analyzer to measure the magnitude of the transmission versus wavelength [23]. Another approach uses a combination of a tunable laser and a photodetector to scan the input wavelength and read the output intensity, respectively [1,24,25]. However, another compact and cost-effective approach recently proposed by Chrostowski et al. [1] replaces off-chip tunable lasers with a chip-integrated fixed wavelength laser. In-resonator phase shifters [26] are used to tune the resonance, and the transmission is read out using a photodetector.

Three metrics that are particularly valuable for evaluating SiP sensor performance and comparing different resonator architectures are the bulk sensitivity, *S_b_*, quality factor, *Q*, and intrinsic limit of detection, *iLoD* [21,27]. The bulk sensitivity is defined as the change in λ*_res_* for a one unit change in the bulk refractive index [27]:(2)$$S_{b}=\frac{\Delta \lambda_{res}}{\Delta n_{bulk}}=\frac{\lambda_{res}}{n_{g}}\left(\frac{\partial n_{eff}}{\partial n_{bulk}}\right),$$
where *n_g_* is the group index and *∂**n_eff_*/*∂**n_bulk_* is the index susceptibility and relates to the portion of the optical mode that interacts with the analyte [21]. Experimentally, *S_b_* can be obtained by exposing the sensor to several solutions having different known refractive indices and tracking the corresponding resonance peak shifts. Often, aqueous solutions prepared with different concentrations of salt [19], isopropanol [3], or glycerol [28,29] are used. *S_b_* is then calculated from the slope of the resonance peak shifts plotted against the bulk refractive index.

The quality factor is a dimensionless quantity that represents a photon’s lifetime in the resonator and is the number of oscillations required for the photon’s energy to decay to 1/*e* [21,22,27]. A high quality factor indicates that light present in the resonator interacts with the analyte for a greater amount of time, and is desirable because it improves the resolution to which the resonance peak shifts can be resolved and reduces the impact of the intensity noise on the resolved shifts [21,22,27,30,31]. The quality factor depends on the total distributed optical losses in the resonator, *α* (dB/m), and can be calculated according to Equation (3) [27].
(3)$$Q_{intrinsic}=\frac{2\pi n_{g}\cdot 10log_{10}\left(e\right)}{\lambda_{res}\cdot \alpha },$$

For MRRs, light must be coupled out of the resonator to observe a resonance change, which degrades the quality factor. In the critically coupled condition, the quality factor is degraded by half compared with the intrinsic quality factor represented by Equation (3), because the proportion of the power coupled out of the resonator is equal to the round-trip loss, effectively doubling the total lost power with each resonator round trip [21]. As such, the critically coupled quality factor is a more useful metric for MRR sensors. Experimentally, it can be approximated based on the full width at half maximum (*FWHM*) at resonance (Δ*λ_FWHM_*) according to Equation (4) [21,22].
(4)$$Q_{crit}=\frac{\pi n_{g}\cdot 10log_{10}\left(e\right)}{\lambda_{res}\cdot \alpha }=\frac{\lambda_{res}}{\Delta \lambda_{FWHM}},$$

Finally, the *iLoD* is a figure of merit introduced to objectively compare sensors, independent of their experimental setups, functionalization strategies, and assays [21,22,27,32]. Unlike the system limit of detection (*sLoD* (RIU)) [21] or analyte limits of detection (M or g/mL) [22], it depends only on the intrinsic characteristics of the resonator and represents the minimum refractive index unit change (in RIU) required to shift the resonance wavelength by Δ*λ_FWHM_*. It is given by Equation (5) [21,22].
(5)$$iLoD=\frac{\lambda_{res}}{Q\cdot S_{b}},$$

Accordingly, *S_b_* and *Q* should be maximized for optimal sensor performance. Resonators designed with conventional strip waveguides (Figure 1c) operating in the quasi-transverse electric mode (hereafter referred to as the TE mode for brevity) can achieve very low optical losses, and therefore, high quality factors [33]. However, the high index contrast between the silicon waveguide and cladding material (typically aqueous solutions for biosensing applications) results in strong confinement of the electric field in the waveguide core. This results in little overlap between the evanescent field and analyte, limiting *S_b_* [3]. Different waveguide designs have been investigated to achieve higher sensitivities, including thin strip waveguides [34], strip waveguides operating in the quasi-transverse magnetic (TM) mode (hereafter referred to as the TM mode for brevity) [21,35], and slot waveguides [36,37].

Sub-wavelength grating (SWG) waveguides (Figure 1d) are yet another geometry that has demonstrated considerable sensitivity enhancements compared to strip waveguides operating with both TE and TM polarizations [38]. SWGs are periodic structures that consist of silicon blocks, interspaced with a lower refractive index material, such as the cladding material (e.g., air [39], water [6,19,40], or a polymer such as SU8 [41]). SWG structures significantly extend the SiP design space by allowing for the fabrication of metamaterial anisotropic structures using standard single-etch CMOS-compatible techniques [42]. SWGs have been used to create photonic structures with tailored modal confinement, broadband behavior, dispersion control, and polarization management [42,43]. For example, the tailorability of modal confinement in SWGs has allowed for the design of ultralow loss waveguide crossings [41] and efficient couplers to interface on-chip waveguides with off-chip optical fibers [44]. The tailorable modal confinement and diffraction suppression afforded by SWGs have been employed to design ultracompact and broadband Y-branches [45] and adiabatic couplers [46,47]. Further, the controlled dispersion of SWGs has been leveraged to design broadband 2 × 2 interferometric switching cells [43] and broadband directional couplers [48,49]. Finally, SWG structures have been used to design optimized sensing waveguides [42]. These periodic SWG structures behave as waveguides below the Bragg threshold where the grating period, Λ, is less than half the effective wavelength of light in the waveguide ($\Lambda <<\frac{\lambda }{2n_{eff}}$) [5,38,41,50]. The optical properties (e.g., *n_eff_*_,_
*n_g_*) of the SWG are highly tunable and depend on the waveguide width (*w*), thickness (*t*), and duty cycle (*δ*, the ratio of the silicon block length to the grating period), in addition to the grating period (Figure 1d). Compared with strip waveguides, SWGs can offer reduced electric field confinement in the waveguide core, which increases light interaction with the analyte [6]. As such, MRRs can be fabricated with SWGs for improved bulk refractive index sensitivity *S_b_*. In the literature, many SWG waveguide variants have been used in MRR and racetrack resonator (RTR) sensors, including SWGs that operate in the transverse electric (TE) [19,21,29,39] and transverse magnetic (TM) [51,52] modes, trapezoidal pillar SWGs [53], substrate overetch (SOE) SWGs [6], pedestal SWGs with undercut etching [28], single- and double-slot SWGs [54,55], and multibox SWGs [3].

When used for biosensing applications, a major limitation of these SWG waveguides, which are composed of isolated silicon blocks, is that they are fragile and susceptible to damage during and after manufacturing [1]. In contrast to other SWG devices (e.g., waveguide crossings, Y-branches, adiabatic couplers, and directional couplers) which are often clad with silicon dioxide, an oxide-open etch is typically necessary to expose SWG-based sensors to the analyte solution [56]. This oxide-open etch can cause delamination of the fragile silicon blocks that make up the SWG waveguides from the sensor substrate [1]. The exposed SWG waveguides can also be damaged during surface functionalization processes and binding assays [57,58,59]. This hinders the fabricability and robustness of SWG-based biosensors, complicating their translation to scalable POC sensors. One solution is to add a fishbone to the SWG waveguide (Figure 1e), which turns the waveguide into a single piece of silicon [1,60,61,62]. This lowers the risk of delamination and improves fabricability, while maintaining the sensitivity enhancements offered by the SWG design. As an additional advantage, the fishbone eliminates discontinuities in tapers which convert routing waveguides to the SWG bus region, reducing reflections and optical losses at the taper interface [1]. To our knowledge, only two other works have reported the design and fabrication of sensors based on fishbone SWG structures [1,62]. Bickford et al. [62] designed Mach-Zehnder interferometers based on fishbone SWG waveguides and presented their transmission spectra, but did not quantify device performance in terms of *Q*, *S_b_*, or *iLoD*. Chrostowski et al. [1] designed a resonator with integrated photoconductive waveguide heater detectors for operation with a fixed-wavelength laser. A fraction of the resonator consisted of a fishbone SWG waveguide. This device achieved an experimental quality factor of 4.44 × 10^4^ and a simulated bulk sensitivity of 76.0 nm/RIU, yielding an estimated *iLoD* of 3.77 × 10^−4^ RIU. No existing works have reported the experimental sensitivity of fishbone SWG sensors. Moreover, to the best of our knowledge, no previous works have demonstrated a comprehensive optimization of the fishbone SWG waveguide geometry for sensing in terms of duty cycle, fishbone width, and grating period.

In this work, we present a novel framework for using numerical simulations to optimize fishbone SWG waveguides for high sensitivity MRRs, aiming to achieve comparable sensing performance to previously reported MRRs based on non-fishbone SWG waveguides, but with improved robustness. For the first time, we demonstrate the experimental performance of MRRs entirely fabricated with fishbone SWG waveguides and compare them to boneless SWG MRRs in terms of the key sensor performance metrics *Q_crit_* (hereafter, simply referred to as *Q*), *S_b_*, and *iLoD*. While the full function of a biosensor depends on several factors beyond the transducer itself, including the functionalization chemistry and assay design, characterizing the intrinsic resonator performance based on these metrics is essential to drawing fair comparisons to other transducers. In both simulations and experiments, we target sensor operation in the O-band (1260–1360 nm) and C-band (1530–1565 nm). While most SiP applications use C-band light, the O-band offers lower optical losses due to reduced water absorption, which has the potential to enhance sensor performance by improving *Q* [21]. To our knowledge, this is the first demonstration of any SWG ring resonators using O-band light for liquid-phase sensing. This is a valuable contribution in the context of POC biosensing, as compact O-band lasers are less expensive and easier to manufacture than C-band ones [1,63], making O-band systems more suitable for affordable and scalable SiP biosensing platforms. Lastly, we compare the performance of our fishbone SWG MRRs to other SWG sensors reported in the literature. This thorough optimization and experimental characterization of fishbone SWG MRRs is an important step toward designing sensitive SiP biosensing platforms that are practical for the POC. In the future, we envision that these sensors can be used for robust biosensing in applications such as the detection of cancer [14,15,16,17,64], inflammation [65], cardiac disorders [66,67], viral infection [68], bacteria [69], and toxins [70,71].

## 2. Materials and Methods

### 2.1. Numerical Models

#### 2.1.1. Index and Bulk Sensitivity Simulations

Finite difference time domain (FDTD) simulations were performed using FDTD Solutions from Lumerical (Ansys, Inc., Canonsburg, PA, USA). In these simulations, one unit cell of a SWG waveguide was modeled with Bloch boundary conditions. The periodicity of SWG waveguides permits the use of band structure simulations for reduced simulation time compared to discrete time domain calculations. This method has been widely used to simulate structures such as photonic crystals [19,72]. In this method, light is injected into the structure over the frequency range of interest and the time-dependent response of the structure is recorded for a range of swept wavevector values [73]. Spectral analysis is performed on this response by searching for local maxima and plotting them in the frequency domain to provide the band structure. Using linear regression to fit the band structure curve, the ratio of the angular velocity to the wavevector is obtained as well as higher order terms. This helps extract the phase and group velocities (*v_p_* and *v_g_*, respectively) from which the effective and group indices are calculated, according to Equations (6)–(8) [74,75],
(6)$$\omega =\omega_{0}+k_{x}\;\frac{\partial \omega }{\partial k_{x}}+\ldots ,$$
(7)$$n_{eff}=\frac{c}{v_{p}}=\frac{c}{\omega /k_{x}},$$
(8)$$n_{g}=\frac{c}{v_{g}}=\frac{c}{\partial \omega /\partial k_{x}},$$
where *ω* is the angular frequency, *k_x_* is the wavenumber, and *c* is the speed of light in a vacuum. To set up these simulations, the silicon SWG waveguide was drawn on top of a 2 µm-thick SiO_2_ buried oxide (BOX) layer with a silicon wafer layer beneath (Figure 2). Water was used as the background (cladding) material. Multi-coefficient material models based on empirical complex refractive index data available from the Lumerical Material Database were used for the simulations [76,77]. The software’s default material model fitting parameters were used for silicon and SiO_2_. As Lumerical’s default fitting parameters yielded an unsatisfying fit for the complex refractive index of water over the O- to C-band wavelength range (Appendix A), the fit tolerance for this material was reduced to 1 × 10^−6^ with the maximum coefficients parameter set to 10 [78]. To better capture the absorption losses of the water cladding, the imaginary weight was increased to 100. This meant that the fitting routine gave 100 times more consideration to the imaginary part of the complex refractive index than the real part; increasing the imaginary weight is recommended when the imaginary refractive index is much smaller than the real refractive index [78]. This produced a model that accurately fit the empirical refractive index data (Appendix A). Light was set to propagate along the x-axis. The FDTD simulation region enclosed one unit cell of the SWG waveguide in the x-direction and extended 0.75 µm above and below the waveguide in the z-direction and 3 µm on either side of the waveguide in the y-direction. These boundary locations were selected based on convergence testing. Bloch boundary conditions were used for the x boundaries. Perfectly matched layers (PML) were used for the z boundaries and one of the y boundaries to absorb waves propagating outwards and avoid reflections, whereas an anti-symmetric condition was used for the other y boundary to reduce the simulation time. The global mesh accuracy was set to 4 and an override mesh (*dx* = 0.01 µm, *dy* = 0.02 µm, *dz* = 0.02 µm) was included in the FDTD region immediately around the waveguide (dimensions defined by Λ × *w* × *t*). A plane wave source was used to inject light into the structure over a frequency range of 120–270 THz (corresponding to a wavelength range of 1111–2500 nm) to cover the O-C band spectra. A band structure analysis group was set up in the FDTD region with ten time monitors randomly distributed in the waveguide.

The effective index and group index versus wavelength were then calculated by sweeping *k_x_* across ten evenly spaced values within a specified range. For a given SWG waveguide geometry, this range was defined by firstly running a coarse sweep with a *k_x_* range of 0.1–0.5 in order to extract the *k_x_* values that corresponded to *n_eff_* at 1310 nm and 1550 nm according to $k_{x}=n_{eff}\Lambda /\lambda$. These values helped define a narrower simulation range with an added buffer of 0.02, which ran with a finer 10-point sweep.

#### 2.1.2. Propagation Loss Simulations

A similar band structure FDTD simulation method to that described in Section 2.1.1 was used to estimate the propagation losses of the SWG waveguides [72]. For these simulations, however, a dipole cloud light source was used to inject light into the structure over a 1-THz frequency range about the operating frequency. For operation in the C-band at 1550 nm, a frequency range of 193.05–194.05 THz (corresponding to a wavelength range of 1546–1554 nm) was used, whereas for operation in the O-band at 1310 nm, a frequency range of 228.51–229.51 THz (corresponding to a wavelength range of 1307–1313 nm) was used. A field decay analysis group was added to the simulation, which included two time monitors placed at different points along the waveguide. The field decay along the waveguide, captured by the time monitors, and the group velocity, obtained from the FDTD simulations described in Section 2.1.1, were used to calculate the propagation loss, *α* (dB/m), according to Equation (9):(9)$$\alpha =\frac{\beta \cdot 20\cdot log_{10}\left(e\right)}{v_{g}},$$
where *β* (Np/s) is the slope of the field decay over time obtained from the simulation ($1\;Np=20\cdot \log_{10}\left(e\right)\;\mathrm{dB}$) [79]. In these loss simulations, the z-span of the FDTD region and override mesh were extended to 3 µm above and below the waveguide. This reduced the risk of losses to the PML boundaries and extended the simulation region into the silicon wafer below the BOX to account for optical losses due to leakage to the substrate. As these simulations were less time-consuming than the sweeps described in Section 2.1.1, the global mesh accuracy was increased to 6 and the override mesh accuracy was increased (*dx*, *dy*, *dz* = 0.01 µm) to improve the simulation accuracy. For each SWG geometry, the loss simulations were performed using the *k_x_* value corresponding to the effective index of the structure simulated in Section 2.1.1.

### 2.2. Design and Optimization of Fishbone SWG Waveguides

In order to optimize fishbone SWG waveguides for sensing applications and compare their performance to conventional boneless SWG waveguides, we performed fully vectorial 3D-FDTD band structure simulations using Bloch boundary conditions, as described in Section 2.1.1. These simulations were used to predict the effective index, *n_eff_*, and bulk sensitivity, *S_b_*, of SWG waveguides operating with C-band and O-band light in the TE mode. Compared to *S_b_*, surface sensitivity (*S_s_*) is the more important metric for biosensors in the study of target molecule quantification, but it must be defined for a specific molecule of interest, meaning that *S_b_* is a more suitable criterion for the general comparison of sensors when the target is unknown or the sensors are used for different biosensing assays [3,40]. As such, *S_b_* was used in this work to compare sensing architectures. For all simulations, a waveguide width of 500 nm, waveguide thickness of 220 nm, and BOX thickness of 2 µm, were used. The grating period, Λ, was initially fixed at 250 nm. This grating period was selected, as it is below the Bragg threshold (Λ << λ/2*n_eff_*) for all studied geometries. Further, others [19,39] have studied boneless SWG waveguides with this grating period, providing a valuable benchmark for comparison. The waveguides were optimized by performing simulation sweeps in which the duty cycle, *δ*, was varied from 0.2 to 0.8 for SWGs with fishbone widths, *w_fb_*, of 0, 60, 100, 140, 180, and 220 nm. Simulations performed with water cladding were used to extract *n_eff_* and the group index, *n_g_*, for each waveguide geometry. To extract *S_b_*, band structure simulations were additionally performed using an index-shifted water cladding material to simulate a dilute salt solution. For this index-shifted material, the real part of the refractive index of water was shifted by 0.01 (Δ*n_bulk_*) at all wavelengths in the water material model; it was assumed that material absorption, and therefore, the imaginary term of the refractive index, remained constant. By simulating *n_eff_* in both materials to extract Δ*n_eff_*, the susceptibility, ∂*n_eff_*/∂*n_bulk_*, could be estimated as Δ*n_eff_*/Δ*n_bulk_*. Using this susceptibility alongside the group index, *S_b_* was calculated according to Equation (2).

Figure 3 presents the results of these simulations. Increasing *δ* and *w_fb_* led to an increase in *n_eff_* for the C-band and O-band structures. This reflects an increase in light confinement as the volume fraction of silicon in the SWG structure increases. This increased light confinement decreases the interaction of light with the bulk material. As seen in Figure 3, this is generally accompanied by a decrease in *S_b_*. However, when *n_eff_* approaches and falls below ~1.44, which is the refractive index of the BOX, the waveguide no longer effectively guides light, and a considerable decrease in *S_b_* is observed when *δ* and *w_fb_* are decreased further [74]. For the C-band devices, the greatest value of *S_b_* out of all the simulated structures was roughly 470 nm/RIU, whereas that for the O-band devices was roughly 400 nm/RIU. The greater sensitivities of the C-band structures can be attributed to lower mode confinement at longer wavelengths at the defined waveguide geometry of *w* = 500 nm and *t* = 220 nm [20].

The sensitivity results highlight that fishbone SWG waveguides can achieve comparable sensitivities compared with boneless SWG waveguides for appropriate combinations of *δ* and *w_fb_*. For both fishbone and boneless SWG structures, the electric field is highly concentrated in the gaps between the silicon blocks, as shown in Figure 4. This allows for strong interaction between the evanescent field and the bulk medium.

Next, to investigate the effect of Λ on the waveguide performance, band structure simulations were performed in which the duty cycle was varied from 0.2 to 0.8 for SWGs with Λ = 200, 250, and 290 nm. These simulations were performed with *w_fb_* = 0 and 100 nm to analyze the effect of Λ on both conventional and fishbone SWGs. The results of these simulations are presented in Figure 5. Note that O-band simulation results are not presented for the fishbone waveguide at *δ* = 0.7 and 0.8 for Λ = 290, nor are they presented for the boneless waveguide at *δ* = 0.8 for Λ = 290, as these structures exceed the Bragg threshold. For the C-band devices, *n_eff_* and *S_b_* are nearly constant across all three values of Λ for a given *δ* and *w_fb_*. Similarly, for the O-band devices, Λ had a small effect on *n_eff_* and *S_b_*, however a small increase in *n_eff_* is seen with increasing Λ, particularly for waveguides approaching the Bragg threshold. Nevertheless, below the Bragg limit, the effect of Λ on the simulated waveguide performance is much less pronounced than the effect of *δ* and *w_fb_*. This is consistent with observations regarding the accuracy of the equivalent refractive index method in predicting SWG behavior well below the Bragg threshold [38,50,81]. The equivalent refractive index method approximates the SWG as a homogeneous strip waveguide with an equivalent refractive index, *n_eq_*, given by Rytov’s formula, $n_{eq}^{2}\approx \delta n_{Si}^{2}+\left(1-\delta \right)n_{clad}^{2}$, where *n_Si_* and *n_clad_* are the refractive indices of the silicon blocks and the cladding material, respectively [38,50]. Using this method, less computationally taxing 2D simulations can be used to estimate the optical properties of the waveguide (e.g., *n_eff_*, *n_g_*, *S_b_*, and *α*) [19]. It has been reported that this method provides suitable approximations for SWG structures in the deep-SWG regime, which is well below the Bragg threshold [37,41]. As *n_eq_* is independent of Λ for any given *δ*, in this regime, the waveguide’s optical properties are, therefore, relatively insensitive to Λ. However, the accuracy of this model degrades near the Bragg threshold, and accurate analysis of the waveguide requires 3D analysis of the periodic geometry and the propagating Bloch–Floquet modes [19,50]. Therefore, near the Bragg threshold, it can no longer be assumed that *n_eff_* and *S_b_* are independent of Λ, which supports the results illustrated in Figure 5.

Based on this analysis, we selected two C-band and two O-band fishbone SWG waveguide designs for fabrication. Given the small effect of Λ on waveguide performance, we chose devices with Λ = 250 nm. Three evaluation criteria were used to select the best combinations of *δ* and *w_fb_* for the fabricated structures. First, the minimum feature size had to exceed 60 nm, which was the minimum fabricable feature size of the ANT electron-beam foundry process used in this work [82]. Next, the reduced modal confinement of SWG waveguides can lead to considerable optical losses to the substrate [3,80]. Sarmiento-Merenguel et al. reported that these substrate leakage losses are independent of SWG geometry and established a direct relationship between leakage losses and *n_eff_*, along with practical design guidelines [80]. In particular, for a 2 µm BOX layer, for C-band light, substrate leakage losses are negligible when *n_eff_* > 1.65. Therefore, in this work, only fishbone SWG designs with simulated *n_eff_* values above this cutoff were considered for fabrication. It should be noted that this leakage loss cutoff was only previously validated for a wavelength range of 1.5–1.6 µm [80]. The leakage loss cutoff is expected to be lower for the O-band than the C-band due to the higher modal confinement at lower wavelengths [20], making 1.65 a conservative estimate for this wavelength range. A comprehensive investigation of O-band substrate leakage losses, although beyond the scope of this work, would validate this assumption and establish a more precise substrate leakage loss cutoff for the O-band. As such, in this work, we used the same leakage loss cutoff of 1.65 for both the C-band and O-band devices. Lastly, among the fishbone SWG designs that satisfied the first two selection criteria, the two C-band and two O-band devices with the highest values of *S_b_* were selected. When selecting the optimized C-band devices, an exception was made, as the geometry with the greatest *S_b_* (*δ* = 0.6 and *w_fb_* = 60 nm) only exceeded the *n_eff_* leakage loss cutoff by ~0.02. To mitigate the risk of leakage losses due to smaller-than-predicted feature sizes, we selected the C-band waveguide geometries with the second- and third-greatest simulated *S_b_* values. The selected C-band (C1 and C2) and O-band (O1 and O2) designs, along with their simulated *n_eff_* values, are provided in Table 1.

In addition to these optimized fishbone SWG designs, an additional six fishbone and boneless SWG waveguides (C3–C6 and O3–O4) with similar *n_eff_* values to the optimized designs were included on the fabricated photonic chips. Their geometries and simulated *n_eff_* values are provided in Table 1. These additional geometries were included to experimentally investigate variations between ring resonators fabricated with fishbone SWGs and conventional SWGs, and to experimentally investigate the effect of grating period on device performance.

### 2.3. Sensor Chip Design and Fabrication

The SWG MRR photonic circuits were designed using KLayout mask editing software, the open-source SiEPIC tools library, SiEPIC EBeam process design kit, and Applied Nanotools process design kit [82,83,84]. One half of the chip layout was dedicated to the C-band resonators, whereas the other half was dedicated to the O-band resonators. All fabricated resonator designs are included in Table 1. The layout included input and output grating couplers to couple light between the chip and benchtop tunable lasers and detectors. 500 nm-wide strip routing waveguides were used to transmit C-band light between the I/Os and resonators, whereas 350 nm-wide strip waveguides were used for the O-band routing. Waveguide bends were designed with a bend radius of 5.0 µm and a Bezier bend parameter of 0.2 [85]. 15 µm-long tapers were used to create smooth transitions between the routing waveguides and the SWG bus regions of the resonators.

The photonic chips were fabricated on silicon-on-insulator (SOI) wafers by Applied Nanotools Inc. (Edmonton, AB, Canada) using 100 keV electron beam lithography and reactive ion etching [82]. All waveguides and photonic structures consisted of silicon. The chips were fabricated with a 220 nm silicon device layer, comprising the patterned photonic circuit, on top of a 2.0 µm SiO_2_ buried oxide (BOX) layer, on top of a 725 µm silicon wafer layer. For this work, the chips were fabricated without cladding. No photoresist or hard mask remained on the waveguide surfaces after fabrication. The chips were used as received for testing. The water contact angle of the sensor chips was found to be 28–30°, representing the hydrophilicity of the BOX layer, which comprises most of the chip’s surface area. It is possible, however, that the silicon waveguides with native oxide exhibit different wetting behavior [86].

### 2.4. Sensor Characterization

The photonic sensors’ transmission spectra were measured to characterize their performance in terms of *n_g_*, free spectral range (*FSR*), extinction ratio, and *Q*. These measurements were made using a custom optical testing setup (Maple Leaf Photonics, Seattle, WA, USA) mounted on a pneumatic vibration isolation table (Newport Corporation, Irvine, CA, USA). The photonic chip was placed on a motorized XY stage (Corvus Eco, Micos GmbH, Eschbach, Germany), maintained at 22 °C with a thermoelectric cooler controlled by a laser diode controller (Stanford Research Systems LDC500, Sunnyvale, CA, USA) and illuminated by a cold light illumination source (Hund, Wetzlar, Germany). A 12-channel lidless fiber array (VGA-12-127-8-A-14.4-5.0-1.03-P-1550-8/125-3A-1-1-0.5-GL-NoLid-Horizontal, OZ Optics, Ottawa, ON, Canada) mounted to a motorized Z stage was aligned to the on-chip grating coupler inputs and outputs. Alignment was performed using open-source PyOptomip software (Python 2.7, 32-bit) [87], which controlled the position of the XY and Z stages and communicated with the tunable lasers and detectors. The relative positions of the photonic chip and fiber array were monitored using top- and side-view microscope cameras (Pixelink, Ottawa, ON, Canada) mounted to 12× zoom lenses (Navitar, Ottawa, ON, Canada). To test the C-band devices, the fiber array was connected to an Agilent 8164A mainframe (Agilent Technologies, Inc. Santa Clara, CA, USA) with a C-band swept tunable laser (Agilent 81682A); to test the O-band devices, the fiber array was connected to another Agilent 8164A mainframe with an O-band swept tunable laser (Agilent 81672B). Eight fiber array channels were connected to Agilent 81635A and Keysight N7744C (Keysight Technologies, Santa Rosa, CA, USA) optical detectors; therefore, up to eight resonators could be probed simultaneously. PyOptomip software was used to control and interface with the tunable lasers and optical detectors.

Prior to the measurements, the resonators were pipette-spotted with ~20 µL of ultrapure water from a NANOpure water purification system (Thermo Fisher Scientific Inc., Waltham, MA, USA). Measurements were then performed by sweeping the tunable laser input and recording the transmission spectra of the resonators. All of the SWG MRR sensors were characterized on five replicate chips.

To extract the sensor performance criteria from the optical spectra, a custom semi-automated script was written in MATLAB (MathWorks, Natick, MA, USA). First, the user was presented with a plot of the overlaid optical spectra of the simultaneously measured 8 resonators and prompted to select the wavelength range to be analyzed. On each optical spectrum, the script then performed (1) peak-finding (findpeaks() function) to identify resonance peak positions and approximate peak widths, (2) fitting of the baseline (non-peak) regions of the spectra to a third-degree polynomial function (polyfit()) and subtraction of that baseline from the optical spectra, (3) linearization of the decibel-scale baseline-subtracted data, (4) nonlinear least-squares fitting of each resonance peak to a Lorentzian function (lorentzfit() 1.7.0.0 by Jered Wells on the MATLAB File Exchange). During step (1), peaks of interest were automatically distinguished from noise by setting the arguments passed to findpeaks() based on the expected form of the data. Specifically, the minimum peak prominence (height of the peak, or extinction ratio) was set to 2 dBm, and the minimum distance between neighboring peaks (FSR) was set to 2 nm. The script also plotted and saved figures highlighting the found peaks on the optical spectra so that the user could check for anomalous results during or after analysis. The fit was performed on the linearized, baseline-subtracted data, and the peak was inverted and normalized prior to Lorentzian fitting (the fitted peak was positive and extended from 0 to 1). If the goodness-of-fit was sufficiently high (R^2^ > 0.85), the center wavelength of the Lorentzian function was used as the resonance peak position in subsequent computations, and the peak’s *FWHM* was calculated from the Lorentzian fit. If the goodness-of-fit was insufficient, the raw peak location was used as the resonance peak position and the *FWHM* was not computed (the peak was not counted in subsequent quality factor analysis). The peak prominence from the peak-finding function was taken as each peak’s extinction ratio, the *FSR* was calculated as the average distance between the resonance peaks in the spectrum (and *n_g_* was computed from the *FSR* as $n_{g}=\frac{\lambda^{2}}{L\cdot FSR}$ [52]), and the quality factor was calculated from Equation (4) using the *FWHM* extracted from the Lorentzian fit.

### 2.5. Microfluidic Design and Fabrication

Microfluidic gaskets to deliver aqueous solutions for sensor performance characterization were fabricated using Sylgard™ 184 poly(dimethylsiloxane) (PDMS) (Ellsworth Adhesives, Hamilton, ON, Canada) molded against 3D printed molds using soft lithography. 2D layouts of the microfluidic channel and mold geometry were designed using KLayout mask editing software (aligned with the photonic design in the same layout), and the microfluidic layers of the layout (separate layers for the outside of the mold, the interior mold cavity, the channel features, and the input/output through holes) were exported as a .dxf file which was subsequently imported into SolidWorks (Dassault Systèmes, Vélizy-Villacoublay, France) and extruded into the final 3D geometry of the mold. The mold created gaskets with two parallel microfluidic channels, each designed to be 200 μm in width and 200 μm in height over the region of the photonic chip containing the sensors, expanding into 500 μm diameter circular input/output regions. The inset region of the mold into which PDMS was cast was designed to be 4 mm in thickness, and the mold also contained 500 μm diameter circular through-hole features to serve as input/output ports. All through-hole features were extruded to a 0.1 mm taller height than the walls of the mold to ensure that thin PDMS membranes did not remain atop through-hole features (the results of experimental testing suggested that 0.1 mm additional height was sufficient to create effective through-holes, whereas 0 mm height differential was insufficient). The gasket mold also contained 3 mm diameter through hole features to self-align the gaskets to the photonic chip, with the chip positioned in a precision-machined recess in a custom-made aluminum mounting plate with matched 4–40 tapped bolt holes. The cast gasket was designed to have ~3.3 mm of extra PDMS on the long edge closest to the channels to reduce any demolding-related feature distortion. This extra PDMS was manually cut off of the fabricated gasket using a single-edge razor blade after demolding.

The molds were printed on a ProFluidics 285D digital light processing (DLP) 3D printer (CADworks3D, Toronto, ON, Canada) at 50 μm using Master Mold resin (CADworks3D). Standard post-processing (isopropanol wash, compressed air dry, and 40 min ultraviolet cure in a Creative CADworks CureZone UV curing chamber (CADworks3D)) was performed on the molds to prepare for soft lithography. The root-mean-squared roughness of the fabricated molds had an upper bound of approximately 65 nm [88]. No mold release agent was used. Sylgard™ 184 silicone elastomer prepolymer base and curing agent (Ellsworth Adhesives, Hamilton, ON, Canada) were mixed at a 10:1 ratio by hand-stirring and a planetary centrifugal mixer (THINKY ARE-310, THINKY USA, Laguna Hills, CA, USA), cast in the 3D printed molds (slightly overfilling the mold so that the PDMS liquid surface was convex and approximately 1 mm above the top of the mold), and degassed in a vacuum desiccator for 30–60 minutes. A sheet of overhead projector transparency material (Apollo, ACCO Brands Corporation, Lake Zurich, IL, USA) was cut to ~4 × 7 cm in size and slowly and carefully laid upon the mold, starting from one corner, to reduce the incidence of bubbles between the PDMS and transparency film [88]. A piece of 1/8”-thick acrylic was then placed atop the transparency and a weight (~500–1000 g) was placed on the acrylic to press the stack together and remove residual PDMS prepolymer between the through-hole features and the transparency film. The use of the transparency and weight system during fabrication produces flat gaskets with complete through holes. The gaskets were cured overnight at 65 °C in an oven (Fisher Isotemp^®^ Incubator 255D, Thermo Fisher Scientific, Hampton, NH, USA), the transparency film was carefully peeled off, and the gasket was then demolded and cut to size. After inspection with optical microscopy (Aven MicroVue Digital Microscope, Aven Tools, Ann Arbor, MI, USA), the gasket was ready for assembly with the photonic chip and mounting plate.

To assemble the setup for fluidic testing (Figure 6), the photonic chip was first placed in the machined recess of the mounting plate. A rectangular washer of the same dimensions as the fluidic gasket and with 4.5 × 2.5 mm rectangular holes aligned with the fluidic I/Os was custom laser-cut from ⅛” acrylic (McMaster-Carr, Elmhurst, IL, USA) using a Universal Laser Systems VersaLaser VLS2.30 laser cutter (Universal Laser Systems, Inc., Scottsdale, AZ, USA). 4–40 brass bolts (McMaster-Carr, Elmhurst, IL, USA) were threaded through the bolt holes in the acrylic washer (first, so that the washer sat against the bolt head) and the PDMS fluidic gasket to align the two pieces together. The bolts were then aligned with the threaded holes in the mounting plate and screwed into place to align and seal the fluidics against the photonic chip. The washer serves to provide even pressure to the flat PDMS gasket to maintain a good seal without a permanent plasma bond between the PDMS and the photonic chip.

### 2.6. Bulk Sensitivity Testing

Bulk sensitivity measurements were performed by measuring the resonance wavelength shifts of the SWG MRRs during exposure to NaCl (Fisher Scientific S271-3, Thermo Fisher Scientific, Hampton, NH, USA) solutions with five different salt concentrations (0 M, 0.0625 M, 0.125 M, 0.250 M, and 0.375 M) and known refractive indices. The solutions were prepared using ultra-pure water. The refractive indices of the solutions were measured with an Abbe refractometer (Spectronic Instruments, Inc., Rochester, NY, USA). From lowest to highest concentration, the measured refractive indices of the solutions were 1.3335, 1.3341, 1.3346, 1.3360, and 1.3373. It should be noted, however, that these are visible wavelength refractive indices and do not account for chromatic dispersion.

The photonic chip was assembled with the microfluidic gasket and mounting plate, as described in Section 2.5. To perform the bulk refractive index sensing measurements, the photonic chip assembly was secured on the stage of the custom optical testing setup (Maple Leaf Photonics, Seattle, WA, USA) using thermally conductive tape. A Fluigent LineUp™ series fluid control system (Fluigent, Le Kremlin_Bicêtre, France) was used to supply fluid to the photonic chip assembly. Further details about this setup are provided in Appendix A.

Prior to the experiment, all tubing in the Fluigent system was primed with fluid by flowing the NaCl solutions at 500 mbar from each of the five reservoirs of both channels in sequence for two minutes each. The primed PEEK tubing (Idex 1531B, Cole-Parmer Canada, Quebec, QC, Canada) connected to the bubble trap outlets (Diba Omnifit^®^ #006BT-HF, Cole-Parmer Canada, Quebec, QC, Canada) was then inserted into the microfluidic inlets of the PDMS gasket. Water was flowed continuously through the microfluidic channels at 10 µL/min until the beginning of the bulk refractive index sensing experiment. The fiber array, connected to the tunable laser and optical detectors, was aligned to the photonic chip, as described in Section 2.4.

During the experiment, the salt solutions were flowed over the MRR sensor via the two microfluidic channels in sequence at 30 µL/min for 20 minutes each. In the first replicate of the experiment, the salt solutions were flowed over the MRR sensor in order of ascending concentration, starting with water (0 M NaCl), followed by 0.0625 M, 0.125 M, 0.250 M, and lastly 0.375 M NaCl solutions. In the second replicate, the salt solutions were flowed over the MRR sensor in order of descending concentration. This was repeated four more times to reach a total of ten replicates. It is important to note that a known limitation of PDMS is that it can leach uncured oligomers into microchannels, with the oligomer concentration being inversely proportional to the flow rate [89]. Given the relatively high flow rate of 30 µL/min used in this study (corresponding to a residence time of ~2 s in the microchannels), in addition to the considerable precedent for use of PDMS-based microfluidics in SiP assays [3,28,71,90], oligomer leaching was expected to have a negligible effect on the bulk refractive index sensing experiments performed in this work. PDMS is also known to absorb small hydrophobic molecules, with absorption increasing with increasing residence time [91,92]. While not a concern in this study, which only used aqueous salt solutions and short residence times, this would be a relevant consideration in sensing assays using longer residence times and precious, low-concentration, and hydrophobic samples.

During the experiment, a custom Python acquisition script was used to sweep the tunable laser source over a 20 nm wavelength range (1540–1560 nm for the C-band devices and 1290–1310 nm for the O-band devices) and record the output transmission spectra from the photonic chip every 20–30 s. The fiber array alignment was monitored and adjusted every 30 sweeps using a fine align function to ensure good coupling to the on-chip grating couplers throughout the experiment.

Acquired optical spectra were analyzed using a custom Python script to Lorentzian-fit each resonance peak and track the cumulative peak shifts, generating plots and datasets of average resonance peak shift vs. time for each measured microring resonator sensor. Briefly, the custom Python script identified resonance peaks in the optical spectra and fit each resonance peak to a 4-parameter Lorentzian function (x-position of the peak center, height of the peak baseline, height of the peak, and peak width at vertical midpoint). It thus parameterized each resonance peak into a 4-element vector and each optical spectrum with n resonance peaks as an n × 4 matrix. It then matched resonance peaks in consecutively acquired spectra by computing the cosine similarity of the vectors [93], and computed the differential displacement in the x-position of the peak centers of the matched peaks. Finally, it averaged the computed differential displacement of all of the matched resonance peaks in the spectra to calculate the overall differential displacement for that sweep iteration (δλ). The overall resonance peak shift at time point *i* ($\Delta \lambda \left(t_{i}\right)$) was calculated as the sum of all preceding displacements: $\Delta \lambda \left(t_{i}\right)=\overset{i}{\underset{1}{\sum }}\delta \lambda$.

All resonances demonstrated a gradual blue drift throughout these experiments. Therefore, prior to further analysis, the peak shift data were drift-corrected by performing a linear fit to the baseline of each peak shift plot and subtracting this linear fit from the data. From the resonance peak shift vs. time data, the bulk refractive index sensitivity values were computed using a custom MATLAB script. The MATLAB script plotted the resonance peak shift vs. time data and prompted the user to click on the regions of the plot corresponding resonator response to each bulk refractive index standard saline solution. For each refractive index standard region, the script averaged the resonance peak shift data in a 20-timepoint region (corresponding to approximately 400s of acquisition) centered at the user’s click location. The bulk refractive index difference was computed as the difference between the measured refractive index of each refractive index standard saline solution and that of water. It then performed a linear regression on the peak shift vs. measured bulk refractive index difference (forcing zero intercept), and the slope of the linear regression was taken to be the bulk refractive index sensitivity.

### 2.7. SEM Imaging

A Zeiss Sigma scanning electron microscope (SEM, Carl Zeiss AG, Jena, Germany) was used to image the fabricated photonic chips. Imaging was carried out to compare the designed dimensions to the fabricated structures and identify any fabrication limitations or unexpected effects. In-lens and secondary electron detectors were used to take top-view and angled-view (45° tilt) images of the photonic devices. ImageJ was used to measure the dimensions of the fabricated SWG waveguides on top-view SEM images taken at 50,000× magnification. For each geometrical parameter (*w*, Λ, *δ*, and *w_fb_*), five measurements were taken and then averaged to give a more representative estimation.

## 3. Results

### 3.1. Simulation Overestimates In-Water Group Indices of SWG Waveguides

Silicon microring resonators with the waveguide geometries outlined in Table 1 were fabricated on a SOI wafer with no oxide cladding using ANT’s electron-beam lithography process [82]. A circular ring geometry was used for the sensors instead of a racetrack geometry to eliminate mode-mismatch losses [3]. All microrings were designed with a radius, *R*, of 30 µm, which was selected to ensure low bend losses [3,94]. To characterize the fabricated microring resonators, a tunable laser was coupled to the devices and their transmission spectra were collected while sweeping the wavelength of the input laser from 1530–1560 nm for the C-band devices or 1270–1310 nm for the O-band devices. This characterization was performed with a droplet of water fully covering the regions of the chip containing the resonators. The measurements were performed on five replicate chips and the measured spectra were analyzed using a custom script, as described in Section 2.4.

Table 2 reports the simulated and measured group indices and *FSR*s of the fabricated ring resonators. All measured group indices were lower than those predicted by simulations, with the boneless SWG devices generally exhibiting a slightly greater difference in *n_g_* between the measured and simulated values compared with the fishbone devices. Accordingly, the measured *FSR*s were greater than the simulated values for all geometries.

Our group has previously fabricated boneless SWG microring resonators using the identical geometry as design C6 from this work, using a different electron-beam lithography fabrication process [19]. Previously, 30 µm-radius ring resonators fabricated with this waveguide geometry (Λ = 250 nm, *δ* = 0.7, *w* = 500 nm, *t* = 220 nm) exhibited an experimental *n_g_* of 3.27 and *FSR* of 3.936 nm, which align well with the simulated values reported here. This indicates that simulation inaccuracies are unlikely to be the source of variation in *n_g_* and *FSR* between the simulated and measured results. Instead, these variations are likely attributable to experimental factors, such as differences between the designed and fabricated structures. In particular, we hypothesized that the low experimental group indices may be due to smaller-than-designed feature sizes on the fabricated chips. To test this hypothesis, SEM imaging was performed on the fabricated structures and feature sizes were measured.

The SEM imaging highlighted two unexpected observations regarding the fabricated waveguide morphologies. First, regarding fabrication tolerances, most waveguide features were slightly smaller than designed. Typically, *w* was 18–25 nm smaller than designed, *w_fb_* was 0–11 nm smaller than designed, and *δ* was approximately equal to the designs. Corners were slightly rounded, though this effect was small. Second, boneless SWG design O4, which had the smallest silicon pillars out of the fabricated devices, showed many collapsed pillars, as seen in Figure 7c. Stiction is known to cause damage to features on micro- and nanoscale devices when exposed to liquid, then dried [58]. Capillary forces pull the feature toward the substrate or adjacent features during this process, leading to deformation, and inhibiting reuse of the device [57,58,59]. The SEM-imaged chip shown in Figure 7 had been exposed to water for characterization prior to imaging, meaning stiction is a likely cause of the visualized damage. Subsequent SEM imaging of an unused chip suggested that this damage was not present before exposure to liquid. This type of damage was not observed for any other waveguides, including design O3, which had the same geometry as O4, but with the addition of a 100 nm fishbone, highlighting the additional structural stability conferred by the fishbone. Drying SiP chips in a low-surface tension solvent (e.g., pentane) is one strategy employed by foundries to prevent stiction during the fabrication process [59,95]. In many applications, however, this may not be feasible due to incompatibilities between these solvents and microfluidic materials [58]. Additionally, the chemical processes used to functionalize SiP sensors for biosensing often involve aqueous solutions and expose the chip to multiple cycles of wetting and drying [96]. In these applications, fishbone SWGs can reduce the risk of damage prior to biosensing assays.

To assess the effect of fabrication tolerance on device performance and determine if the low experimental group indices could be attributed to the smaller-than-designed feature sizes observed in the SEM images, we re-ran band structure simulations for designs C1 and C4, this time using their measured geometries. The smoothing of corners was not included in these simulations due to the small magnitude of this effect, as observed in SEM images. However, this corner smoothing should be accounted for in simulation models of waveguides fabricated with photolithography techniques (e.g., Deep UV lithography), which are known to cause prominent corner smoothing [83]. The group indices obtained from these simulations were 3.108 and 3.110, respectively, which represent a ~3–7% reduction in *n_g_* compared with the original simulations, but not a sufficiently large reduction to completely account for the experimental results.

Another possible explanation for the low experimental group indices is incomplete wetting of the SWG structures. Nanostructured surfaces can be susceptible to this phenomenon, which leads to the entrapment of air between narrow features during wetting [97]. As such, air may have been trapped between the silicon pillars when the photonic chips were coated with water for measurements. Because air has a lower refractive index than water, this is expected to decrease *n_eff_* [50]. The group index can be related to *n_eff_* according to $n_{g}\left(\lambda \right)=n_{eff}\left(\lambda \right)-\lambda \cdot \left(dn_{eff}/d\lambda \right)$ [74], where $dn_{eff}/d\lambda <0$ for the designed waveguides. While the first term of this equation should decrease in the case of incomplete wetting, the magnitude of the second term should also decrease when air is added to the SWG metamaterial, as air is less dispersive than water [77]. Depending on the relative effect of trapped air on these two terms, incomplete wetting may cause a decrease in *n_g_*. To theoretically test this hypothesis, simulations were performed with fishbone SWG design C1 in which the gaps between the silicon pillars were filled with air up to a height *t_air_* (Appendix A). The fabricated waveguide geometry, as measured from SEM images, was used. Further details regarding these simulations are provided in the Appendix A. These simulations showed that the combination of reduced feature sizes and air entrapment considerably reduced *n_g_* and an increase in *t_air_* led to a decrease in *n_g_* (Appendix A). An air pocket height of *t_air_* = 120 nm yielded *n_g_* = 2.825, which is very close to the experimentally measured value of *n_g_* = 2.83. It should be noted that this model does not account for the curvature of the air–water interfaces enclosing the air pocket. Regardless, these simulation results suggest that the low experimental *n_g_* values may, indeed, be the result of incomplete wetting. Similarly to stiction during drying, incomplete wetting can cause deformations and damage to structures adjacent to the trapped air due to capillary pressure [97]. This may have contributed to the feature collapse seen in Figure 7c. Another similar phenomenon that may have contributed to the low group indices is nanobubble formation on the waveguide surfaces due to etch roughness [98]. The presence of a thin native oxide layer on the waveguide surface is yet another factor that may have contributed to these results [99].

### 3.2. Empirical Characterization of Extinction Ratio vs. Coupling Gap Reveals Insights for Further Optimization and Highlights Performance Degradation Due to Peak Splitting

Critical coupling is achieved when the coupling gap, *g_c_*, between the bus waveguide and ring resonator is such that the power coupled into a ring resonator is equal to the round-trip losses in the ring [20]. At critical coupling, the extinction ratios (ERs) of the resonance peaks are maximized, thus enhancing the signal-to-noise ratio; this is a desirable condition for robust peak tracking and sensitive analyte detection [98]. When *g_c_* is relatively small, the resonator is over-coupled, giving rise to increased power losses. This decreases both ER and *Q*. When *g_c_* is relatively large, the resonator is under-coupled, which increases *Q*, but decreases ER. Indeed, under-coupling can be used to enhance *iLoD*, although a tradeoff with ER exists for noisy systems that necessitate higher ERs for robust peak tracking [100]. In this work, we aimed to optimize ER to facilitate straightforward extraction of the sensor intrinsic quality factor for comparison with propagation loss simulations, as well as facilitate meaningful comparison to previously reported sensors operating near critical coupling [3,21,28]. Subsequent system design (building upon the optimization framework presented here) should consider the tradeoff between *Q* and ER in choosing the best coupling condition for the application, and may choose to under-couple the resonators.

To achieve critical coupling, *g_c_* can be selected based on numerical simulations. For example, the critical coupling condition can be estimated based on simulated coupling coefficients extracted from FDTD simulations of the entire coupling region, along with simulated propagation losses [20,101]. However, one drawback of this approach is that FDTD simulations of the coupling region are very computationally intensive for SWG resonators. Additionally, while these FDTD coupling coefficient and loss simulations account for loss contributions due to material absorption and substrate leakage, they often do not accurately recapitulate the effects of optical scattering, which depend on the surface roughness of the fabricated waveguides and can increase losses and affect the coupling condition [20]. Scattering has an increased effect on SWG waveguides compared to conventional strip waveguides owing to the increased surface area of SWG structures [3,39]. Considering these limitations, we decided to take an empirical approach to optimize *g_c_* for close-to-critical coupling.

Each resonator was fabricated with four different coupling gaps. The fabricated coupling gaps for the C-band devices were based on our group’s previous empirical findings for conventional SWG ring resonators with similar expected effective indices. As outlined in Table 1, coupling gaps of *g_c_* = 450, 500, 550, and 600 nm were fabricated for devices C1, C2, C4, and C5, which had simulated effective indices between 1.70–1.71. Smaller coupling gaps of *g_c_* = 400, 450, 500, and 550 nm were selected for C3 and C6 due to their greater predicted effective indices and, therefore, increased optical confinement. It has been reported that coupling increases with increasing wavelengths due to reduced optical confinement at the defined waveguide geometry of *w* = 500 nm and *t* = 220 nm [20]. As such, smaller coupling gaps were selected for the O-band devices, relative to their predicted effective indices. Coupling gaps of *g_c_* = 400, 450, 500, and 550 nm were fabricated for O1 and O4, whereas coupling gaps of *g_c_* = 350, 400, 450, and 500 nm were fabricated for O2 and O3 due to their higher simulated effective indices.

The extinction ratios for all resonator designs were measured, as described in Section 2.4, and the results are presented in Figure 8 and Table 3. This characterization was performed for five replicate chips and mean values are reported. Details regarding the number of resonance peaks included from each chip in each mean calculation are provided in Appendix A. As shown in Figure 8b, all C-band devices, excluding C3, exhibited maximum extinction ratios at their largest coupling gaps. Consequently, it cannot be concluded that critical coupling was achieved for these devices, and future work should include the fabrication of these resonators with larger coupling gaps to avoid over coupling. In SEM images, the measured coupling gaps were 20–40 nm smaller than designed, which may be related to proximity effect correction in the lithography process [102]. This may have contributed to this requirement for larger coupling gaps. As illustrated in Figure 8c, devices O1, O2 and O3 exhibited maximum extinction ratios at intermediate values of *g_c_* within their fabricated ranges. However, the variations in extinction ratio between different values of *g_c_* are similar in magnitude to the standard deviations of the measurements, so these results may not confirm critical coupling. Resonator O4 achieved an extinction ratio at its largest fabricated coupling gap, further highlighting that future work should extend the coupling gap ranges investigated here.

Variations in maximum extinction ratios between the devices, in particular between the C-band and O-band devices, may be attributable to peak splitting. Peak splitting was visible in the resonator spectra and was particularly prominent for the O-band devices. This peak splitting, which is discussed further in the next section, leads to deleterious effects on the resonator performance, including a reduction in peak height, potentially explaining why the maximum extinction ratios measured for the O-band resonators were lower than those measured for the C-band resonators [103,104]. Peak splitting largely arises due to stochastic scattering effects, which vary with wavelength [20]. This can lead to unpredictability in peak splitting severity between resonances, which may account for the large standard deviations of the measured extinction ratios [20]. This peak splitting may also be responsible for the absence of prominent maxima for devices O1, O2 and O3 in Figure 8c. As the measurements were made over wavelength ranges of 30–40 nm, it is likely that the wavelength-dependence of coupling within these wavelength ranges also contributed to the large standard deviations [20]. Finally, the detectors used in the experimental characterization of the sensors had a minimum detectable power of −80 dBm, which meant that some high-extinction ratio peaks were clipped at their minima. This may have added to the large standard deviations and may have caused an underestimation of some extinction ratios for designs close to critical coupling.

### 3.3. Fishbone SWG MRRs Achieve Comparable Performance to Previously Reported SWG-Based Sensors

Quality factors were estimated for all fabricated resonator designs by simulating the waveguide propagation losses, as described in Section 2.1.2, then calculating the critically coupled quality factor, according to Equation (4). The simulated losses and corresponding quality factors for all waveguide designs are presented in Table 4. Indeed, these simulated losses and quality factors do not account for the effects of fabrication-related optical losses, but they do provide the fundamental limit for the device performance [3]. This offers a valuable benchmark against which to compare experimental results, which can help to identify the contribution of fabrication-related losses to the real device performance and inform future approaches to mitigate these effects.

Optical absorption in water is the dominant loss mechanism for waveguides operating in the C-band. Since the predicted effective indices were similar for all of the fabricated C-band designs, indicating similar modal confinement, similar losses were expected among these devices [3]. The simulated losses aligned well with this, as all simulated losses were between 39.9–40.7 dB/cm, suggesting that *w_fb_* and *δ* have little effect on the material losses of SWG waveguides with similar effective indices.

It has been reported that the optical absorption of water is roughly ten times lower in the O-band than the C-band, allowing for significantly lower material losses [3,21]. This is reflected in the simulated propagation losses for the O-band structures, which ranged from 6.1–7.5 dB/cm. These losses are roughly six times lower than the C-band propagation losses, which is a smaller reduction in losses compared to what would be predicted if the losses were solely due to material absorption. This discrepancy may be due to small losses to the substrate and PML boundaries. Corresponding to their lower losses, the simulated quality factors for the O-band resonators were considerably greater than those for the C-band resonators, highlighting the potential benefit of using the O-band light for sensing applications.

Quality factors for the fabricated ring resonators were calculated from the measured spectra, as described in Section 2.4, and the results are provided in Table 4. For the C-band devices, the simulated quality factors were 1.3–1.6 times as large as the experimental values. This difference between simulated and experimental values is likely due to scattering and coupling losses, which were not accounted for in the simulations. Scattering losses arise due to roughness introduced on the waveguide surfaces during fabrication, which makes them challenging to model. These losses are typically non-negligible for SWG waveguides owing to their large surface area [3,94]. Next, overcoupling leads to greater optical losses compared to critical coupling [100]. As discussed in the previous section, many of the C-band resonators were likely overcoupled, giving rise to this loss mechanism. Since the simulated quality factors were calculated based on the critical coupling assumption, these losses are another likely source of variation between the simulated and experimental results. It should be noted that the propagation loss simulations described in this work also did not include bending losses. Based on previously reported results, we expected negligible bending losses at the large ring radius of 30 µm considered here [3].

The simulated quality factors for the O-band resonators ranged from 4.40 × 10^4^ to 5.11 × 10^4^, whereas the experimentally measured values were 6.3–7.2 times lower (Table 4). While scattering and coupling losses, combined with the smaller-than-designed feature sizes of the fabricated structures, likely contributed to this discrepancy, peak splitting appeared to be the dominant source of this variation. In an ideal ring resonator, there exist two counterpropagating circulating modes, clockwise and counterclockwise, which are uncoupled, and degenerate, meaning they resonate at the same frequency [104,105]. In this case, the resonator exhibits single peaks. A small mode perturbation, however, can couple these modes and break their degeneracy leading to a resonance shift that manifests as split resonance peaks [20,103,104]. In silicon waveguides, this perturbation typically occurs due to stochastic backscattering arising from sidewall roughness [20,103,104]. In the spectra measured for all O-band resonators, peak splitting was prevalent, comprising 18–51% of all resonances. Conversely, split peaks were far less common in the C-band resonator spectra, comprising roughly 2–12% of all resonances. While all resonators studied in this work were fabricated using the same foundry process and, therefore, were subject to similar sidewall corrugations, the exaggerated peak splitting observed among the O-band devices suggests that sidewall scattering is exacerbated at lower wavelengths. This is consistent with analytical models for scattering losses in which the losses are proportional to the square of the ratio of surface roughness to the wavelength of light in the material [106]. Thus, the effects of scattering, and therefore, peak splitting, increase with decreasing wavelength. Additionally, the higher water absorption at 1550 nm may be hiding peak splitting, whereas a 10× lower water absorption at 1310 nm would reveal scattering induced peak splitting.

The analysis script used to extract the quality factors from the measured spectra performed Lorentzian fitting on the resonance peaks to measure the *FWHM*, from which the quality factors were calculated. In the case of split peaks, the Lorentzian was typically fit to the doublet, leading to an underestimation of *Q* (Figure 9). In this analysis, a R^2^ cutoff of 0.85 dictated which peaks were used in the calculation of *Q*. The split peaks typically exhibited poor R^2^ values compared with single peaks (Figure 9b); however, there was considerable overlap, with some apparent split peaks exhibiting higher R^2^ values than some single peaks (Figure 9c). However, it should be noted that some peaks, such as the one shown in Figure 9c, exhibited apparent peak splitting that had a similar magnitude to the spectral noise, making it challenging to confidently confirm the identity of these peaks as split or non-split.

To test whether increasing the selection stringency effectively eliminated split peaks from the quality factor calculation, the analysis was repeated on the O-band data with the R^2^ threshold increased to 0.95. This analysis did not produce a noticeable difference in the results and incompletely filtered out the split peaks, while eliminating numerous single peaks and underestimating the quality factors. While it may be possible to perform improved fitting to the doublets to extract more accurate quality factors in post processing, their extinction ratios will still be degraded. Further improvements could be made to the analysis algorithm to filter out split peaks and only analyze apparent non-split peaks, but such an approach is confounded by the variable severity of the split peaks. For example, split peaks may be visually imperceptible in cases where the splitting is less than a linewidth, yet these peaks will still exhibit degraded extinction ratios and quality factors. As illustrated in Figure 9c, when the magnitude of peak splitting is similar to the noise in a given spectrum, it may also be challenging to confirm the identity of split peaks with high confidence. Overall, the prevalence of these split peaks is likely to cause deleterious effects in the analysis of binding assays. Therefore, a more robust solution for improving sensor performance is to design resonators that are less sensitive to backscattering.

The back reflections that cause peak splitting have been reported to increase with *n_g_* [20]. This is consistent with our experimental results. Resonator design O4, which had the lowest *n_g_* out of the O-band devices, demonstrated the least peak splitting, with split peaks comprising approximately 18% of all resonances measured across five chips. Resonator designs O2 and O3, which had the two highest values of *n_g_*, demonstrated the most peak splitting, at roughly 50% and 51% of all resonances, respectively. In these estimates, it should be noted that split peaks with very low extinction ratios (e.g., due to overcoupling) were nearly indistinguishable from noise. This meant that some split peaks may have been overlooked, resulting in an underestimation of their true occurrence. Among the resonators that had sufficiently high extinction ratios to confirm the identity of split peaks, O1, O2, O3, and O4 exhibited peak splitting on approximately 42%, 54%, 70%, and 27% of their resonances, respectively. Overall, these data suggest that reducing *n_g_* by reducing *δ* and/or *w_fb_* may improve the resonator performance. This should be accompanied by a detailed analysis of O-band substrate leakage losses to ensure that any feature size reductions do not introduce additional deleterious effects. Finally, electron-beam fabrication processes have been found to yield semi-periodic surface roughness [20]. If the surface roughness of the fabrication process is well-characterized, simulation models can be established to better predict the extent of backscattering at different wavelengths, which may help predict peak splitting and inform ring resonator design [20]. If possible, a reduction in etch roughness could further reduce these scattering effects. Fishbone SWG structures fabricated by Deep UV lithography are likely to have reduced sidewall roughness. In electron beam lithography processes similar to that used in this work, the shot pitch and machine grid are very small (e.g., 5–6 nm) and the electron beam size is roughly 10–20 nm [83]. The quantization of shots, which are not very well smoothed out by the small beam size, to the machine grid results in high-resolution roughness. In contrast, Deep UV lithography processes use masks made by electron beam lithography, but they are smoothed out by the 193 nm wavelength of light used for the exposure and pattern transfer [83]. Hence, peak splitting is expected to be reduced for devices fabricated by 300 mm wafer 193 nm immersion Deep UV lithography foundries. Moreover, Deep UV lithography processes now enable high-volume manufacturing of SiP chips with sub-100 nm feature sizes, making Deep UV lithography an attractive option for mass production of SWG-based sensor chips [42,83,107].

Next, ring resonator performance was assessed in terms of *S_b_*. The simulated *S_b_* values for all fabricated devices are reported in Table 5. To experimentally measure the bulk sensitivities of the fabricated devices, the sensor chips were interfaced with a two-channel PDMS microfluidic gasket and five NaCl solutions with different concentrations (0–0.375 M) were flowed over the sensors in alternating ascending and descending sequences for a total of ten replicate exposures to each solution. Throughout the experiment, the transmission spectra were measured using a tunable laser and optical detectors. The measurement setup used for these experiments is shown in Figure 10a. For each resonator design (C1–C6 and O1–O4), two replicate resonators were monitored, one of which was located in microfluidic channel 1 and the other in channel 2. The same fluid sequence was delivered to both microfluidic channels. On the chip layout, the C-band and O-band devices were accessible from grating couplers on opposite edges of the 9 mm × 9 mm photonic chip. This meant that the microfluidic gasket had to be rotated 180° to access the C-band devices compared to the O-band. When the gasket was aligned to the C-band devices, the O-band resonators were in direct contact with PDMS, and vice versa. As the chips were fabricated without an oxide cladding, this made the resonators in direct contact with the PDMS prone to damage during gasket alignment and removal. Therefore, the C-band and O-band devices were tested with microfluidics on separate chips to prevent damage to the resonators prior to use. The resonators’ saved spectra were analyzed by a custom retrospective analysis script to track the resonance wavelength shifts of the sensors, as described in Section 2.6. An example of a spectrogram collected during one of these experiments and an overlaid peak shift plot generated by the retrospective analysis script are shown in Figure 10b. An example of the spectral peak shifts corresponding to each salt solution is shown in Figure 10c.

The peak shift plots all demonstrated a gradual baseline blue drift over time (10–72 pm/hr). While the source of this drift is unclear, one contributor may be the gradual etching of silicon by NaCl solution [108]. Prior to further analysis, the peak shift data were, therefore, drift-corrected by performing a linear fit to the baseline of each peak shift plot and subtracting this linear fit from the data. *S_b_* was then calculated by performing a linear regression on the resonance peak shifts versus the bulk refractive index of the salt solutions data, and extracting the slope, as illustrated in Figure 11. For each resonator, *S_b_* was calculated as the average slope from 8–10 linear regressions (on the data from 8–10 replicate exposures to all five salt solutions), and the average values and their standard deviations are presented in Table 5. It should be noted that only eight replicates were used when peak shift abnormalities, such as abrupt jumps and drops (likely due to bubbles passing through the fluidic system), were observed during the first replicate. In these cases, the first two replicates were excluded from the calculated averages to maintain an equal number of ascending- and descending-concentration replicates.

On average, the experimental results aligned well with the simulated ones, but the experimental results showed variations in bulk sensitivity between the two microfluidic channels. For the C-band designs, resonators in channel 1 always demonstrated higher values of *S_b_*, whereas for the O-band designs, excluding O1, resonators in channel 2 always demonstrated higher values of *S_b_*. For O1, *S_b_* was virtually identical between the two channels. This spatial variation in *S_b_* is illustrated in Figure 12a. These variations similarly affected the fishbone and boneless structures. To determine if this variation was the result of variability among the fabricated resonator structures, the spatial variation of *n_g_* was similarly analyzed based on the results obtained in Section 3.1 (Figure 12b), which showed that *n_g_* did not vary as a function of location on the chip for replicate resonator designs, though the boneless devices typically exhibited larger differences between the experimental and simulated group indices compared to the fishbone structures. This suggests that experimental factors related to fluidics were the most likely source of variation in *S_b_*.

In these experiments, the NaCl solutions used in both channels were aliquots of the same stock solution, eliminating the solutions as a source of error. Further, the fluid control system was programmed to deliver identical flow rates through both channels. These flow rates were monitored by flow sensors throughout the experiments to ensure that the expected flow rates were delivered, making this another unlikely source of error. The refractive indices of the NaCl solutions used in the *S_b_* calculations were measured with an Abbe refractometer at visible wavelengths and chromatic dispersion was not accounted for, which could constitute one source of experimental error.

The presence of trapped air between the silicon pillars of the SWG waveguides due to incomplete wetting may be responsible for the variations in *S_b_*. Firstly, as discussed in Section 3.1, this phenomenon may have contributed to the lower-than-predicted values of *n_g_*. *S_b_* is inversely proportional to *n_g_*, so a decrease in *n_g_* may lead to an increase in *S_b_*. However, this cannot be decoupled from variations in the susceptibility, which may also arise as a result of trapped air. Trapped air in the gaps between the silicon pillars could reduce the interaction between the evanescent field and bulk fluid in these regions of high electric field intensity, reducing susceptibility. Conversely, the decrease in modal confinement associated with the decrease in *n_eff_* may increase modal overlap with the bulk, potentially increasing susceptibility. Thus, the effects of this phenomenon on *n_g_*, susceptibility, and therefore *S_b_*, are challenging to predict and may account for some of the variability observed here. Appendix A presents simulation results for a model of incomplete wetting for waveguide design C1. As shown in Appendix A, an increase in the height of the trapped air pockets between the SWG pillars decreased *S_b_*. It is possible that differences in the speed at which fluid was introduced to the microfluidic channels affected the extent of wetting and the average sizes of these air pockets. This could have contributed to the differences in *S_b_* between the channels. However, based on the simulation results, these differences in *S_b_* should correlate to differences in *n_g_* between the channels. This was not observed experimentally (Appendix A). Nevertheless, the simulation model does not account for the shape of the air pocket, which may also influence *S_b_*, potentially explaining these differences. Lastly, it should also be noted that the simulation results provided in Appendix A generally predict lower values of *S_b_* compared with the experimental results. These simulated values were obtained at a single wavelength (1550 nm), whereas the experimental measurements were obtained from multiple resonances analyzed over a wavelength range of 1530–1560nm, which likely contributed to these discrepancies between the experiments and simulations.

Overall, the fishbone SWG resonators achieved comparable, and sometimes better, sensitivities than the boneless SWG designs. For the C-band resonators, fishbone device C2 achieved, on average, the greatest bulk sensitivity at 438 and 416 nm/RIU in channels 1 and 2, respectively. For the O-band resonators, boneless SWG design O4 achieved the greatest bulk sensitivity at 364 and 383 nm/RIU in channels 1 and 2, respectively. However, these values only surpassed the best-performing O-band fishbone resonator (O1) by 15 and 34 nm/RIU in channels 1 and 2, respectively.

Figure 13 compares the reported performance of SWG resonators based on quality factor, *S_b_*, and *iLoD*. TE and TM strip waveguide resonators are also included as performance benchmarks. Lines of constant *iLoD* are plotted, showing that *iLoD* decreases and resonator performance improves toward the top right corner of the plot. The C-band and O-band boneless SWG (C6 and O4) and fishbone SWG (C1 and O2) resonators that demonstrated the smallest experimental values of *iLoD* in this work (listed in Table 5) are also included on the plot. The plotted bulk sensitivities for these devices are averages from the two microfluidic channels. It should be highlighted that several of the resonators with the highest *iLoD*s on this plot have been characterized based on simulations with gaseous cladding material (legend entries marked with a section sign, §) [54,55]. These reported levels of performance are likely considerably greater than what can be achieved with the same sensor architectures applied to real-world aqueous-phase sensing due to the additional material and scattering losses. All other sensor performance data are based on experimental results measured with water cladding.

Indeed, this plot highlights that a tradeoff exists between quality factor and *S_b_*, with TE and TM strip waveguide sensors achieving high quality factors, but low sensitivities due to limited light interaction with the analyte. Meanwhile, most SWG designs offer high sensitivities and low quality factors due to increased light interaction with the analyte [1]. This plot illustrates that the fishbone SWG resonators presented in this work have comparable performance to other SWG sensors that have been experimentally demonstrated to date. Given the improved fabricability and structural robustness of fishbone SWGs compared to all other SWG designs, which involve free-standing silicon blocks, these results position fishbone SWG resonators as an attractive architecture for scalable and portable sensing. Moreover, the scattering effects that degraded O-band sensor performance in this work can likely be reduced by fabricating waveguide geometries with slightly lower group indices, designing models to better predict the effects of etch roughness, and using Deep UV lithography fabrication. These improvements could increase the O-band sensor quality factors and push their performance toward simulated values, with the potential for extremely low limits of detection.

## 4. Conclusions

In this work, we demonstrated the optimization and experimental characterization of SiP MRR sensors designed with fishbone SWG waveguides for both O-band and C-band operation. Waveguide designs were optimized based on 3D-FDTD simulations to find combinations of Λ, *δ*, and *w_fb_* that optimize sensitivity while meeting the substrate leakage loss criterion. MRRs were fabricated with the optimized waveguide designs and experimentally tested to evaluate their optical properties, spectral characteristics, and performance compared to boneless SWG MRRs in terms of *n_g_*, *FSR*, extinction ratio, *Q*, *S_b_*, and *iLoD*. The O-band fishbone SWG MRRs achieved quality factors as high as 7.8 × 10^3^, bulk sensitivities as high as 349 nm/RIU, and intrinsic limits of detection as low as 5.1 × 10^−4^ RIU. The C-band fishbone SWG MRRs achieved quality factors as high as 5.5 × 10^3^, bulk sensitivities as high as 438 nm/RIU, and intrinsic limits of detection as low as 7.1 × 10^−4^ RIU. In general, the fishbone SWG resonators presented in this work have comparable performance to other SWG sensors that have been experimentally demonstrated to date, while offering improved fabricability and a lower risk of damage compared with the boneless designs. The performance of the O-band resonators was, however, hindered by peak splitting. This peak splitting was likely the result of scattering effects, which were exaggerated at lower wavelengths and likely exacerbated by fabrication issues. These scattering effects could be reduced by designing waveguides with lower group indices and designing models to better predict the effects of etch roughness. This highlights the potential to realize O-band fishbone SWG MRRs with higher quality factors and lower limits of detection than current state-of-the art SWG sensors. One of the challenges with SWG structures is the small feature size required in fabrication. O-band, as opposed to C-band, involves slightly smaller features, which in this paper were a minimum size of 100 nm, and a minimum gap of 100 nm. These sizes are compatible with deep immersion 193 nm CMOS foundry fabrication, hence can be fabricated in high volume. Overall, the results of this work indicate that fishbone SWG waveguides allow for improved robustness and fabricability without compromising performance. While we have developed a framework for optimizing fishbone SWG MRRs and have experimentally demonstrated their sensing capabilities, the POC use of these transducers relies on system-level integration with other biosensor components. Sensor biofunctionalization, sample delivery, and signal readout strategies must all be optimized for the POC setting in order to successfully translate these SiP devices into fully portable diagnostic tools.

## Figures and Tables

**Figure 1 biosensors-12-00840-f001:**
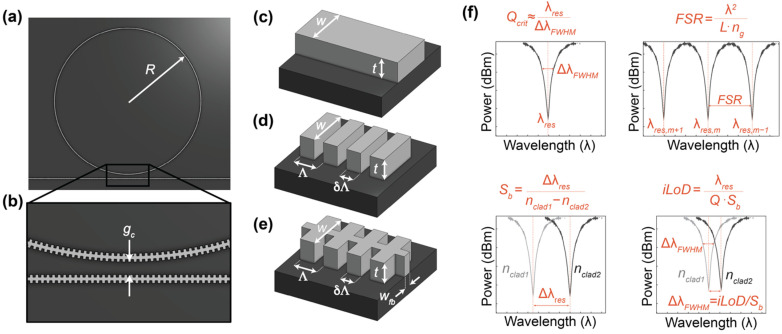
(**a**) Illustration of microring resonator (MRR) with radius *R* fabricated with fishbone sub-wavelength grating (SWG) waveguides and (**b**) expanded view of bus-to-ring coupling region with coupling gap *g_c_*. Illustrations of (**c**) strip, (**d**) conventional SWG, and (**e**) fishbone SWG waveguide geometries with width *w*, thickness *t*, grating period Λ, duty cycle *δ*, and fishbone width *w_fb_*. (**f**) Graphical representations of MRR performance criteria, including (clockwise from top left) critically coupled quality factor (*Q_crit_*), free spectral range (*FSR*), intrinsic limit of detection (*iLoD*), and bulk sensitivity (*S_b_*).

**Figure 2 biosensors-12-00840-f002:**
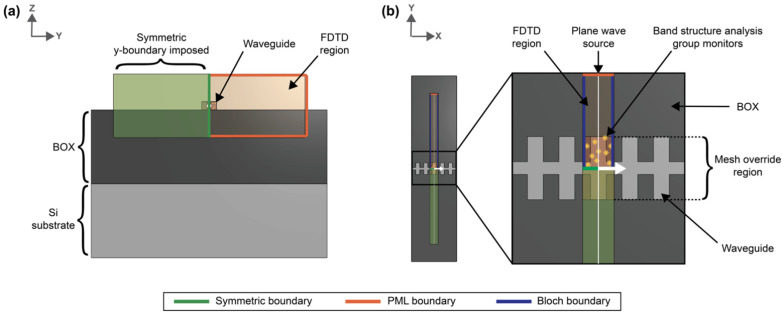
(**a**) Illustration of cross-section of finite difference time domain (FDTD) band structure simulation setup. (**b**) Illustration of top view of FDTD band structure simulation with magnified view of waveguide region.

**Figure 3 biosensors-12-00840-f003:**
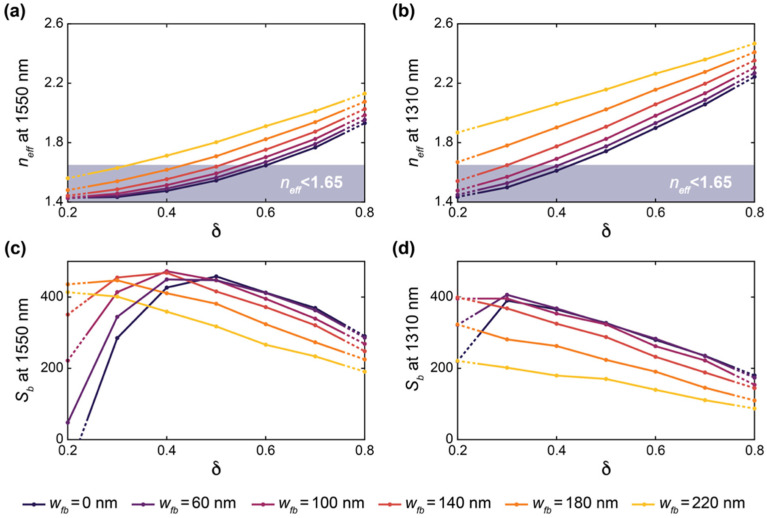
Simulated effective index, *n_eff_*, versus duty cycle, *δ*, for SWG waveguides with a grating period Λ = 250 nm, width *w* = 500 nm, thickness *t* = 220 nm, and fishbone width *w_fb_* = 0–220 nm, operating with (**a**) 1550 nm and (**b**) 1310 nm light. The shaded area represents the region where *n_eff_* is below the substrate leakage loss cutoff of 1.65, as reported by Sarmiento-Merenguel et al. [80]. Dashed regions of curves indicate waveguide geometries with feature sizes smaller than the 60 nm limit of the electron-beam lithography process used for silicon photonic (SiP) chip fabrication in this work. Simulated bulk sensitivity, *S_b_*, versus *δ* for SWG waveguides with Λ = 250 nm, *w* = 500 nm, *t* = 220 nm, and *w_fb_* = 0–220 nm, operating with (**c**) 1550 nm and (**d**) 1310 nm light. Simulations were performed using a 3D-FDTD approach with water as the cladding.

**Figure 4 biosensors-12-00840-f004:**
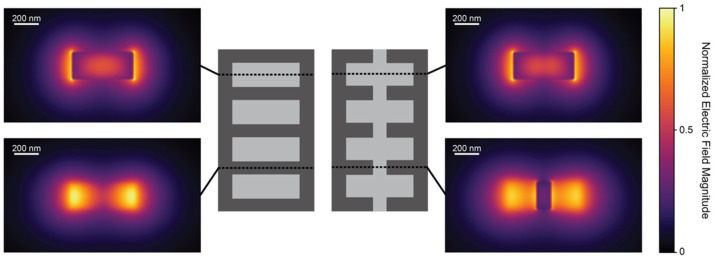
Cross-sectional electric field profiles for boneless (**left**) and fishbone (**right**) SWG waveguides obtained from 3D-FDTD simulations at 1550 nm, illustrating the electric field intensity in the silicon blocks and gaps between these blocks. The boneless SWG field profiles were simulated for a waveguide with the following geometry: Λ = 250 nm, *w* = 500 nm, *t* = 220 nm, *δ* = 0.65, and *w_fb_* = 0 nm (*n_eff_* = 1.70). The fishbone SWG field profiles were simulated for a waveguide with the following geometry: Λ = 250 nm, *w* = 500 nm, *t* = 220 nm, *δ* = 0.60, and *w_fb_* = 100 nm (*n_eff_* = 1.70). The data presented in all subplots have been normalized to the same maximum electric field intensity.

**Figure 5 biosensors-12-00840-f005:**
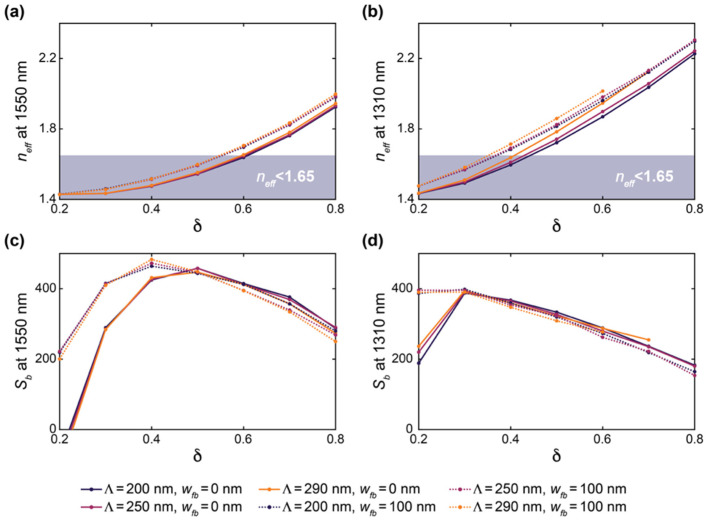
(**a**,**b**) Simulated *n_eff_* versus *δ* for SWG waveguides with Λ = 200, 250 and 290 nm, *w* = 500 nm, *t* = 220 nm, and *w_fb_* = 0 and 100 nm, operating with (**a**) 1550 nm and (**b**) 1310 nm light. The shaded area represents the region where *n_eff_* is below the substrate leakage loss cutoff of 1.65, as reported by Sarmiento-Merenguel et al. [80]. (**c**,**d**) Simulated *S_b_* versus *δ* for SWG waveguides with Λ = 200, 250 and 290 nm, *w* = 500 nm, *t* = 220 nm, and *w_fb_* = 0 and 100 nm, operating with (**c**) 1550 nm and (**d**) 1310 nm light. Simulations were performed using a 3D-FDTD approach with water as the cladding material.

**Figure 6 biosensors-12-00840-f006:**
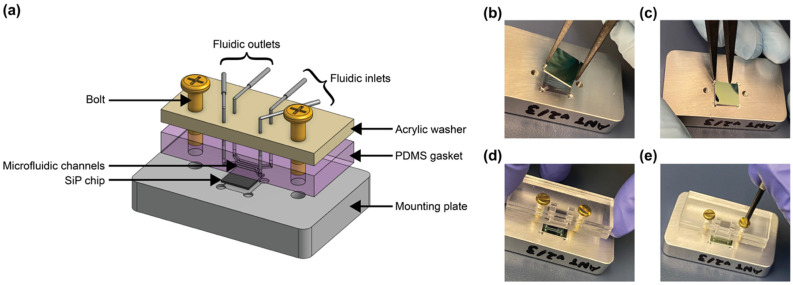
(**a**) Schematic and (**b**–**e**) photos of microfluidic and photonic chip assembly for fluidic testing. (**b**) The chip is placed on the mounting plate, (**c**) the chip is positioned into the machined recess of the mounting plate, (**d**) bolts are threaded through the bolt holes of the acrylic washer and poly(dimethylsiloxane) (PDMS) gasket and aligned to the threaded holes in the mounting plate, and (**e**) bolts are screwed into place to align and seal the fluidics against the photonic chip.

**Figure 7 biosensors-12-00840-f007:**
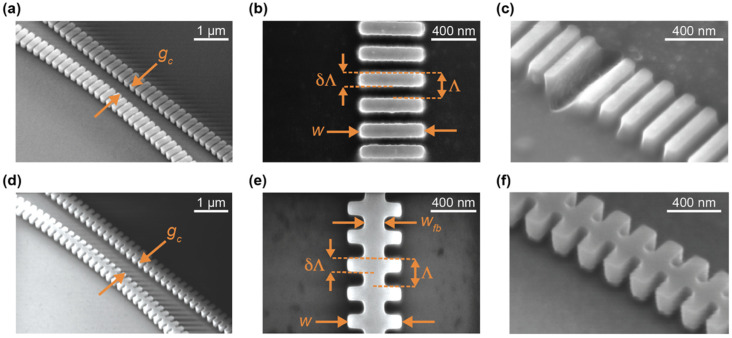
Scanning electron microscope images of fabricated SWG MRRs. (**a**) 45° angled view of coupling region for a boneless SWG MRR, (**b**) top-view of boneless SWG waveguide O4, (**c**) 45° angled view of boneless SWG waveguide O4, showing collapsed silicon pillars after testing, highlighting robustness issues with this boneless design (we did not observe any collapsed pillars in images acquired after fabrication but prior to testing with water), (**d**) 45° angled view of coupling region for a fishbone SWG MRR, (**e**) top-view of fishbone SWG waveguide C1, and (**f**) 45° angled view of fishbone SWG waveguide C2.

**Figure 8 biosensors-12-00840-f008:**
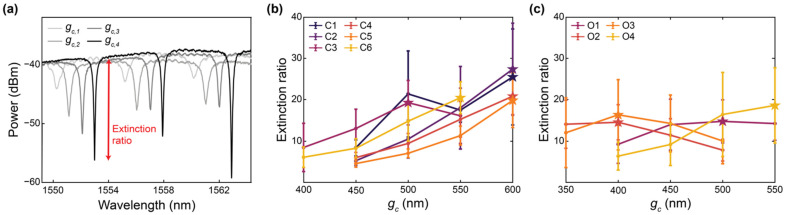
(**a**) Example spectra of four microring resonators with the same ring radii and waveguide geometries but different coupling gaps, leading to variations in resonance peak extinction ratios. Measured extinction ratios for (**b**) C-band and (**c**) O-band microring resonators as a function of coupling gap distance (*g_c_*). Star-shaped markers indicate designs which achieved the greatest extinction ratios for each waveguide geometry. All measurements were performed with water cladding. Extinction ratios for each sensor geometry were calculated from 0–13 resonance peaks per resonator (those peaks passing the R^2^ > 0.85 threshold) analyzed over a wavelength range of 1530–1560 nm for the C-band devices and 1270–1310 nm for the O-band devices, with 1–2 replicate resonators on each of five replicate chips. Extinction ratios are reported as the mean ± one standard deviation. Further details regarding the number of included resonance peaks per resonator on each chip are provided in Appendix A.

**Figure 9 biosensors-12-00840-f009:**
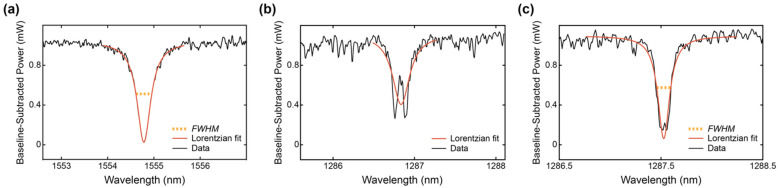
(**a**) Example of a non-split resonance peak from a spectrum measured for resonator C6, showing the Lorentzian fit and full width at half maximum (*FWHM*) calculated by the analysis script. (**b**) Example of a split resonance peak with a low R^2^ (0.794) from a spectrum measured for resonator O3, showing the Lorentzian fit calculated by the analysis script (*FWHM* not calculated, as this peak exhibited a R^2^ value below the cutoff of 0.85). (**c**) Example of an apparent split resonance peak with a high R^2^ (0.951) from a spectrum measured for resonator O3, showing the Lorentzian fit and *FWHM* calculated by the analysis script.

**Figure 10 biosensors-12-00840-f010:**
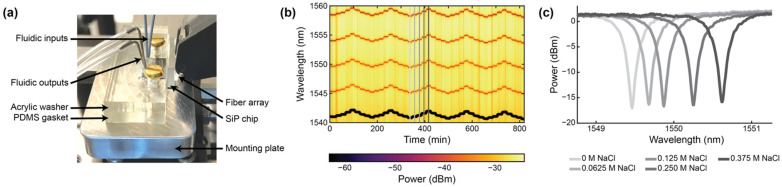
(**a**) Measurement setup for bulk sensitivity experiments, showing a SiP sensor chip interfaced with microfluidics and coupled to a fiber array to connect to benchtop optical inputs and outputs. (**b**) Spectrogram with overlaid peak tracking plot (black line) for device C3 measured during a bulk sensitivity experiment in which the sensor was exposed to five different NaCl solutions, each with different concentrations, a total of ten times. (**c**) Overlaid spectra for device C3 illustrating the red shift in resonance peak wavelength with increasing NaCl concentration. The vertical gray lines in (**b**) indicate the times at which the spectra displayed in (**c**) were obtained during the bulk sensitivity experiment.

**Figure 11 biosensors-12-00840-f011:**
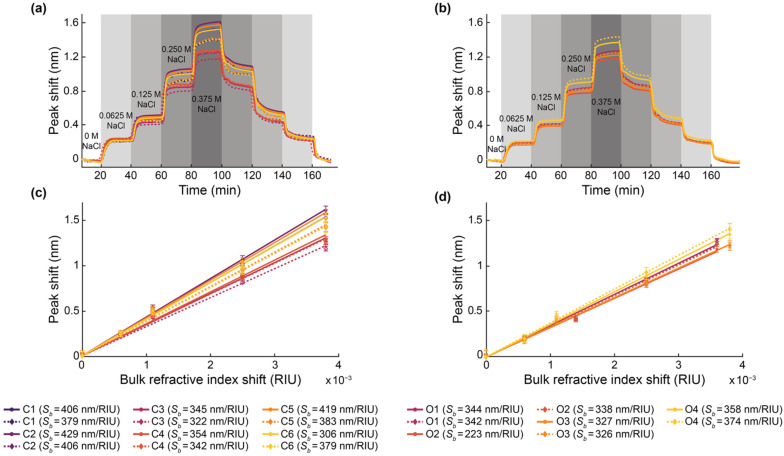
Measured resonance wavelength shifts as a function of NaCl concentration for (**a**) C-band and (**b**) O-band ring resonators. Measurements from the fifth (ascending) and sixth (descending) replicate exposures to five different NaCl concentrations are displayed. Measured resonance wavelength shifts versus bulk refractive index with linear fits for (**c**) C-band and (**d**) O-band ring resonators. The linear fit slopes were used to calculate *S_b_* for each resonator. Note that the results presented in (**c**,**d**) correspond to the fifth replicate of the bulk sensitivity experiments, whereas the *S_b_* values reported in Table 5 represent averages measured across all ten replicates. Solid lines correspond to ring resonators located in microfluidic channel 1, whereas dotted lines correspond to ring resonators located in microfluidic channel 2. Additionally, note that the traces for device C1 are hidden under the traces for device C6 in (**a**,**c**).

**Figure 12 biosensors-12-00840-f012:**
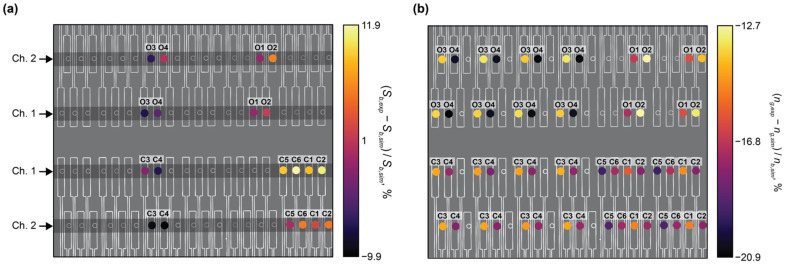
(**a**) Bright-field micrograph of the sensor chip (converted to grayscale) with markers indicating the locations of each ring resonator used for the bulk sensing measurements. The marker colors are mapped to the percent difference between the measured and simulated *S_b_* for each resonator. The positions of the two microfluidic channels for each set of testing (O-band and C-band) is denoted by darker gray rectangular overlays. (**b**) Bright-field micrograph of the sensor chip with markers indicating the locations of each ring resonator characterized in Section 3.1. The marker colors are mapped to the percent difference between the measured and simulated *n_g_* for each resonator.

**Figure 13 biosensors-12-00840-f013:**
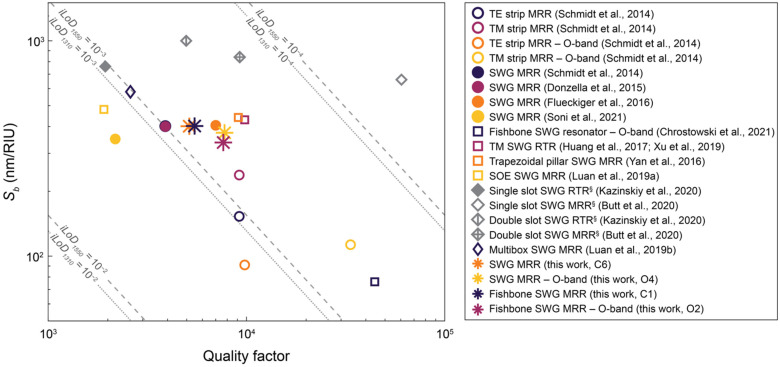
Reported silicon photonic resonator sensors performance comparison in terms of quality factor, *S_b_*, and *iLoD*. All sensors operate in the C-band unless marked otherwise. For works that reported sensor performance data for multiple sensor geometries, the best-performing design has been included in this plot. The sensors include quasi-transverse electric (TE) strip MRR [21], quasi-transverse magnetic (TM) strip MRR [21], TE strip MRR (O-band) [21], TM strip MRR (O-band) [21], SWG MRR [19,21,29,39], fishbone SWG resonator [1], TM SWG racetrack resonator (RTR) [51,52], trapezoidal pillar SWG MRR [53], substrate overetch (SOE) SWG MRR [6], single-slot SWG RTR [54], single-slot SWG MRR [55], double-slot SWG RTR [54], double-slot SWG MRR [55], multibox SWG MRR [3], and the best-performing C-band and O-band boneless (C6 and O4) and fishbone (C1 and O2) SWG MRRs reported in this work (C6 achieved *S_b,ave_* = 401 nm/RIU, *Q* = 5.2 × 10^3^, and *iLoD* = 7.5 × 10^−4^ RIU; O4 achieved *S_b,ave_* = 374 nm/RIU, *Q* = 7.8 × 10^3^, and *iLoD* = 4.5 × 10^−4^ RIU; C1 achieved *S_b,ave_* = 402 nm/RIU, *Q* = 5.5 × 10^3^, and *iLoD* = 7.1 × 10^−4^ RIU; O2 achieved *S_b,ave_* = 337 nm/RIU, *Q* = 7.6 × 10^3^, and *iLoD* = 5.1 × 10^−4^ RIU). Curves of constant *iLoD* are included on the plot for sensors with resonance wavelengths of 1310 and 1550 nm. Legend entries marked with a section sign (§) indicate sensors for which the reported *Q* and *S_b_* were obtained from simulations performed with gaseous cladding material (these markers are depicted in grayscale). Values reported for all other sensors are based on experimental measurements in water cladding.

**Table 1 biosensors-12-00840-t001:** Fabricated SWG ring resonator geometries and their expected effective indices obtained from 3D-FDTD band structure simulations with water cladding. All waveguides have a thickness of 220 nm.

Ring Resonator Design	Λ (nm)	*δ*	*w* (nm)	*w_fb_* (nm)	*n_eff_*	*g_c_* (nm)
C1 ^╫^	250	0.5	500	180	1.71	450, 500, 550, 600
C2 ^╫^	250	0.6	500	100	1.70	450, 500, 550, 600
C3 ^╫^	290	0.65	500	100	1.77	400, 450, 500, 550
C4	290	0.65	500	0	1.71	450, 500, 550, 600
C5	250	0.65	500	0	1.70	450, 500, 550, 600
C6	250	0.7	500	0	1.77	400, 450, 500, 550
O1 ^╫^	250	0.4	500	100	1.69	400, 450, 500, 550
O2 ^╫^	250	0.4	500	140	1.77	350, 400, 450, 500
O3 ^╫^	200	0.5	500	100	1.82	350, 400, 450, 500
O4	200	0.5	500	0	1.72	400, 450, 500, 550

^╫^ Fishbone SWG designs.

**Table 2 biosensors-12-00840-t002:** Group index, *n_g_*, and free spectral range (*FSR*) for SWG ring resonators obtained from 3D-FDTD band structure simulations with as-designed structures and experiments. For each resonator, *n_g_* and *FSR* were calculated based on all resonances measured over a wavelength range of 1530–1560 nm for the C-band devices and 1270–1310 nm for the O-band devices. The reported values are the mean ± one standard deviation for these measurements from five replicate chips.

Ring Resonator Design	*n_g_*		*FSR* (nm)	
	Simulation	Experiment	Simulation	Experiment
C1 ^╫^	3.33	2.83 ± 0.06	3.83	4.51 ± 0.09
C2 ^╫^	3.16	2.60 ± 0.05	4.04	4.91 ± 0.10
C3 ^╫^	3.44	2.94 ± 0.05	3.71	4.33 ± 0.07
C4	3.21	2.64 ± 0.04	3.97	4.82 ± 0.08
C5	3.12	2.52 ± 0.05	4.09	5.05 ± 0.10
C6	3.35	2.77 ± 0.05	3.81	4.61 ± 0.08
O1 ^╫^	3.08	2.57 ± 0.07	2.96	3.54 ± 0.09
O2 ^╫^	3.45	3.00 ± 0.08	2.64	3.04 ± 0.08
O3 ^╫^	3.34	2.89 ± 0.07	2.73	3.15 ± 0.08
O4	3.01	2.39 ± 0.07	3.02	3.80 ± 0.10

^╫^ Fishbone SWG designs.

**Table 3 biosensors-12-00840-t003:** Optimal coupling gaps for each ring resonator geometry and their measured extinction ratios. Extinction ratios for each sensor geometry were calculated from 0–13 resonance peaks per resonator (those peaks passing the R^2^ > 0.85 threshold) analyzed over a wavelength range of 1530–1560 nm for the C-band devices and 1270–1310 nm for the O-band devices, with 1–2 replicate resonators on each of five replicate chips. Extinction ratios are reported as the mean ± one standard deviation. Further details regarding the number of included resonance peaks per resonator on each chip are provided in Appendix A.

Ring Resonator Design	Optimal *g_c_* (nm)	Extinction Ratio (dB)
C1 ^╫^	600 *	26 ± 12
C2 ^╫^	600 *	27 ± 11
C3 ^╫^	500	19 ± 5
C4	600 *	21 ± 4
C5	600 *	20 ± 7
C6	550 *	21 ± 4
O1 ^╫^	500	15 ± 5
O2 ^╫^	400	15 ± 4
O3 ^╫^	400	16 ± 9
O4	550 *	19 ± 9

^╫^ Fishbone SWG designs. * The best-performing coupling gap was the largest out of the fabricated range for a given geometry, indicating potential over coupling.

**Table 4 biosensors-12-00840-t004:** Comparison of SWG ring resonator propagation losses and critically coupled quality factors predicted by 3D-FDTD simulations with experimentally measured quality factors. Simulations and experimental characterization were performed with water cladding. Experimental quality factors are reported for the ring resonator designs with the best-performing coupling gaps, as listed in Table 3. Experimental quality factors for each sensor geometry were calculated from 0–13 resonance peaks per resonator (those peaks passing the R^2^ > 0.85 threshold) analyzed over a wavelength range of 1530–1560 nm for the C-band devices and 1270–1310 nm for the O-band devices, with 1–2 replicate resonators on each of five replicate chips. Experimental quality factors are reported as the mean ± one standard deviation. Further details regarding the number of included resonance peaks per resonator on each chip are provided in Appendix A.

Ring Resonator Design	Simulated Propagation Losses (dB/cm)	*Q*	
		Simulation *	Experiment
C1 ^╫^	40.2	7.29 × 10^3^	(5.5 ± 1.0) × 10^3^
C2 ^╫^	40.7	6.82 × 10^3^	(4.7 ± 0.8) × 10^3^
C3 ^╫^	40.5	7.46 × 10^3^	(4.7 ± 0.5) × 10^3^
C4	40.4	7.00 × 10^3^	(4.9 ± 0.4) × 10^3^
C5	40.1	6.85 × 10^3^	(4.6 ± 0.3) × 10^3^
C6	39.9	7.37 × 10^3^	(5.2 ± 0.5) × 10^3^
O1 ^╫^	7.3	4.40 × 10^4^	(6.7 ± 1.5) × 10^3^
O2 ^╫^	7.5	4.79 × 10^4^	(7.6 ± 1.7) × 10^3^
O3 ^╫^	7.1	4.93 × 10^4^	(6.9 ± 1.6) × 10^3^
O4	6.1	5.11 × 10^4^	(7.8 ± 2.0) × 10^3^

^╫^ Fishbone SWG designs. * Calculated based on critical coupling assumption.

**Table 5 biosensors-12-00840-t005:** SWG ring resonator *S_b_* and *iLoD* values predicted by 3D-FDTD simulations and measured experimentally. Experimental *S_b_* values are reported as averages ± one standard deviation from 8–10 replicate experiments in which the ring resonators were exposed to five NaCl solutions of different concentrations. Experimental *iLoD*s were calculated for each resonator using the experimentally measured *Q* values (Table 4) and the mean of the experimentally measured *S_b_* values from channels 1 and 2, with errors propagated from the standard deviations reported in Table 4 and for *S_b_* in channels 1 and 2.

Ring Resonator Design	*S_b_* (nm/RIU)			*iLoD* (RIU)	
	Simulation	Experiment, Ch. 1	Experiment, Ch. 2	Simulation	Experiment
C1 ^╫^	381	414 ± 8	389 ± 10	5.58 × 10^−4^	(7.1 ± 0.9) × 10^−4^
C2 ^╫^	395	438 ± 8	416 ± 11	5.75 × 10^−4^	(7.7 ± 0.9) × 10^−4^
C3 ^╫^	357	349 ± 6	323 ± 6	5.82 × 10^−4^	(9.8 ± 0.7) × 10^−4^
C4	382	354 ± 5	343 ± 7	5.80 × 10^−4^	(9.1 ± 0.6) × 10^−4^
C5	392	427 ± 8	392 ± 10	5.77 × 10^−4^	(8.3 ± 0.4) × 10^−4^
C6	369	413 ± 8	389 ± 10	5.69 × 10^−4^	(7.5 ± 0.6) × 10^−4^
O1 ^╫^	354	349 ± 11	349 ± 8	8.42 × 10^−5^	(5.6 ± 0.9) × 10^−4^
O2 ^╫^	325	330 ± 15	344 ± 7	8.41 × 10^−5^	(5.1 ± 0.8) × 10^−4^
O3 ^╫^	357	331 ± 6	333 ± 8	8.31 × 10^−5^	(5.8 ± 0.9) × 10^−4^
O4	382	364 ± 7	383 ± 7	7.69 × 10^−5^	(4.5 ± 0.8) × 10^−4^

^╫^ Fishbone SWG designs.

## Data Availability

The data presented in this study are available on request from the corresponding author.

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
