# Peer review of "An Optimization Framework for Silicon Photonic Evanescent-Field Biosensors Using Sub-Wavelength Gratings"

_biosensors, 2022, doi:10.3390/bios12100840_

Round 1

Reviewer 1 Report

The Authors propose a fishbone sub wavelength grating microring resonators. The performance has been estimated by using different software based on 3D-FDTD method. Although the results are promising for the next generation of diagnostic, a major revision of the manuscript is suggested to address the following comments:

-          In the Introduction Section, an overview of the potential applications of subwavelength gratings should be provided (e.g. A review of silicon subwavelength gratings: building break-through devices with anisotropic metamaterials. Nanophotonics, 2021), aiming at highlighting the benefits, as the controlled dispersion (e.g. Design of a large bandwidth 2× 2 interferometric switching cell based on a sub-wavelength grating. Journal of Optics23(8), 085801, 2021).

-          In the Introduction Section, the Authors should discuss about the potential applications and their target performance to help the reader in the device rating. Maybe could be useful to compare the device with other devices reported in literature with the same structure as “Fish-bone subwavelength grating waveguide photonic integrated circuit sensor array. In Chemical, Biological, Radiological, Nuclear, and Explosives (CBRNE) Sensing XIX (Vol. 10629, pp. 109-119). SPIE.” 

-          All tables should be improved, make them more compact and simpler to read. Try to let them be in the same page.

-           The structure of the article is a little bit confusing, the Authors should follow the design steps, then talk about the fabrication and at the end comment results and tests.

Reviewer 2 Report

This manuscript demonstrated a sub-wavelength grating ring resonator with a fishbone structure for refractive index sensing applications. Detailed simulations and experiments are performed to show the effect of multiple parameters. However, I don't think there is enough novelty supporting the publication. Testing only with salt water is far away from a real biosensor. The author should explain more about the innovations and how they compare with other works.

Also, the paper contains too much detail and fundamental knowledge, making it more like a thesis, book chapter or tutorial for fresh students. In my opinion, this paper's length can be reduced to less than 10 pages without losing any key information. I would leave this to the editor to decide whether this is in an acceptable form. If this is what the journal requires, some necessary changes are outlined below.

1.       In line 62, the claim is not accurate. The functionalized surface does not necessarily be a label-based scheme. For example, in [Feng, Xueling, et al. ACS sensors 2.7 (2017): 955-960], [Tao, Jifang, et al. Scientific reports 6.1 (2016): 1-7.], and [Park, Mi Kyoung, et al. 2012 7th ICIEA. IEEE, 2012.]. The author should also compare with the works that use strip waveguide, which is already quite mature and widely used. Compared with SWG waveguide is not convincing enough.

2.       In line 77, would it be better to say "sensitivity to surface and cladding refractive index 'change'"?

3.       In line 205, what does the author mean by "capture the absorption profile of water"? Why is the imaginary weight increased to 100? Does this provide correct results?

4.       In lines 217 and below, could the author consider using wavelength units rather than frequency?

5.       In section 2.1.2, the author also mentioned multiple times that the fabrication-related loss is the main contributor to total propagation loss. Why is the simulation for propagation loss required here if it is not important? Also, if the simulation of propagation loss is not accurate, why the simulation of the Q factor is performed? How can these results be related to the experiment?

6.       In line 240, why not use the dB unit?

7.       In line 295, how to distinguish noise and peaks in the peak-finding functions?

8.       In 2.2 to 2.4, some more important details are not covered. These questions are essential if people want to do surface functionalization. What is the wafer layer structure? What is the waveguide and cladding material? Is there any remaining photoresist and hard mask on top of the waveguide? Is the chip surface hydrophobic or hydrophilic? How to solve the air bubble issue? Does the author have any plan to cover the chip with SiO2? What is the surface roughness of the 3D-printed PDMS mold? How to make sure the channel does not absorb or emit anything that might affect the testing?

9.       In line 679, a higher ER means lower Q as compared to the under-coupled case. This will deteriorate the LOD. How to trade-off the selection of higher ER and higher Q?

10.   In line 690, the author mentioned that the coupling gap is selected based on previous experience. Then why do they need to show the simulation results?

11.   In fig. 7, the axis label is missing.

12.   Why would the author care about peak splitting? Can we just pick one of the best peaks for the sensing application?

13.   In fig. 8, it seems the noise is quite large. How to make sure the result is due to peak splitting rather than just noise?

14.   In lines 876-877, is there any support for this claim? Why does DUV have better sidewall roughness?

15.   In line 968, it doesn't make sense to say that misalignment can cause variation because the evanescent field only extends a few hundred nanometers. Basically, you will see the channel touching the ring at this distance.   

Reviewer 3 Report

The draft titled “An Optimization Framework for Silicon Photonic Evanescent Field Biosensors Using Sub-Wavelength Gratings” described a type of subwavelength grating waveguide, called fishbone, based device for the application of biosensor. After reading through the draft carefully, I was not persuaded to get it published in the current form. The followings are my comments:

1.      The arrangement of the content is a bit confusing. I think the design and optimization parts should be in “Materials and Methods”.

2.      The author described the assembling of the microfluidic and PICs in context, but it would be great if it can be illustrated in figures. (The author only supplies a zoomed out actual photo which is hard to understand the protocol for assembling).

3.      Since the dimension of the subwavelength structures are very small, can the author give a brief analysis on the performance vs fabrication tolerance?

4.      The dimension of the structure is smaller than most of the foundries’ critical dimension (normally 150 or 180nm), can the author comment on the mass-production of the chips in the future for the practical application?

Round 2

Reviewer 1 Report

The Authors have strongly modified the manuscript according to the Reviewer suggestions. However, the main benefits of SWG, as the controlled dispersion (e.g. Design of a large bandwidth 2× 2 interferometric switching cell based on a sub-wavelength grating. Journal of Optics, 23(8), 085801, 2021), should be discussed, to help the reader to rate the proposed device.

Author Response

Point 1: The Authors have strongly modified the manuscript according to the Reviewer suggestions. However, the main benefits of SWG, as the controlled dispersion (e.g. Design of a large bandwidth 2× 2 interferometric switching cell based on a sub-wavelength grating. Journal of Optics, 23(8), 085801, 2021), should be discussed, to help the reader to rate the proposed device.

Response 1: To address this comment, we have modified lines 165-172 of the manuscript to introduce dispersion control as a benefit of SWG structures. We have referred to the paper about interferometric switches by Brunetti et al. (2021) cited by the reviewer [43], in addition to two other papers about directional couplers [48,49], which have leveraged the controlled dispersion of SWGs to design broadband devices.

Corresponding modifications/additions to manuscript for Point 1: (red text represent changes made to the manuscript since the first round of revisions; black text represents text that has been left unchanged since the first round of revisions)

Sub-wavelength grating (SWG) waveguides (Figure 1d) are yet another geometry that has demonstrated considerable sensitivity enhancements compared to strip waveguides operating with both TE and TM polarizations [38]. SWGs are periodic structures that consist of silicon blocks, interspaced with a lower refractive index material, such as the cladding material (e.g., air [39], water [6,19,40], or a polymer like SU8 [41]). SWG structures significantly extend the SiP design space by allowing for the fabrication of metamaterial anisotropic structures using standard single-etch CMOS-compatible techniques [42]. SWGs have been used to create photonic structures with tailored modal confinement, broadband behavior, dispersion control, and polarization management [42,43]. For example, the tailorability of modal confinement in SWGs has allowed for the design of ultralow loss waveguide crossings [41] and efficient couplers to interface on-chip waveguides with off-chip optical fibers [44]. The tailorable modal confinement and diffraction suppression afforded by SWGs have been employed to design ultracompact and broadband Y-branches [45] and adiabatic couplers [46,47]. Further, the controlled dispersion of SWGs has been leveraged to design broadband 2 × 2 interferometric switching cells [43] and broadband directional couplers [48,49]. Finally, SWG structures have been used to design optimized sensing waveguides [42]. These periodic SWG structures behave as waveguides below the Bragg threshold... (lines 157-174)

Reviewer 2 Report

The author addressed all my questions. I'm ok with the publication.

Author Response

No further revisions were requested by this reviewer for this round of minor revisions.

Reviewer 3 Report

The author has answered all of my questions.

Author Response

As per the Review Report, no further revisions were requested by this reviewer for this round of minor revisions.